# One-dimensional single atom arrays on ferroelectric nanosheets for enhanced $CO_2$ photoreduction

Lizhen Liu[1,2,6], Jingcong Hu[3,6], Zhaoyu Ma[4], Zijian Zhu[1], Bin He[1], Fang Chen ®[1] ✉, Yue Lu ®[3] ✉, Rong Xu ®[2], Yihe Zhang[1], Tianyi Ma ®[5], Manling Sui ®[3] & Hongwei Huang ®[1] ✉

Single-atom catalysts show excellent catalytic performance because of their coordination environments and electronic configurations. However, controllable regulation of single-atom permutations still faces challenges. Herein, we demonstrate that a polarization electric field regulates single atom permutations and forms periodic one-dimensional Au single-atom arrays on ferroelectric $Bi_4Ti_3O_{12}$ nanosheets. The Au single-atom arrays greatly lower the Gibbs free energy for $CO_2$ conversion via Au-O=C=O-Au dual-site adsorption compared to that for Au-O=C=O single-site adsorption on Au isolated single atoms. Additionally, the Au single-atom arrays suppress the depolarization of $Bi_4Ti_3O_{12}$, so it maintains a stronger driving force for separation and transfer of photogenerated charges. Thus, $Bi_4Ti_3O_{12}$ with Au single-atom arrays exhibit an efficient CO production rate of 34.15 $\mu mol·g^{-1}·h^{-1}$, ~18 times higher than that of pristine $Bi_4Ti_3O_{12}$. More importantly, the polarization electric field proves to be a general tactic for the syntheses of one-dimensional Pt, Ag, Fe, Co and Ni single-atom arrays on the $Bi_4Ti_3O_{12}$ surface.

$CO_2$ photoreduction as an "artificial photosynthesis" strategy is a promising route to reducing the greenhouse effect and solving the energy crisis[1]. However, the efficiency and product selectivity are far from satisfactory in industry due to the high activation energies. The activities of catalytic sites determine $CO_2$ adsorption and the activation energy barriers of catalysts, which are very important for photocatalysis[2]. Therefore, substantial research has been focused on increasing the activities of catalytic sites.

Single-atom catalysts (SACs) have atomically dispersed catalytic sites with uniform geometric and electronic structures that regulate the surface local charge distribution of the catalyst, providing abundant highly active catalytic sites[3]. Thus, SACs with isolated single atoms (i-SAs) have garnered significant attention in photocatalysis. However, i-SA sites always lack proximity to sites that induce multibond adsorption with reactants, which is often needed in challenging reactions. Recently, SACs with neighboring catalytic sites were found to exhibit dual-atom synergetic catalytic effects unavailable for i-SA sites[4]. Since the electronic and geometric structures of the neighboring single atoms (SAs) resemble those of the i-SAs, they retain the catalytic selectivities of i-SAs while providing higher activities. Prior

[1]Engineering Research Center of Ministry of Education for Geological Carbon Storage and Low Carbon Utilization of Resources, Beijing Key Laboratory of Materials Utilization of Nonmetallic Minerals and Solid Wastes, National Laboratory of Mineral Materials, School of Materials Science and Technology, China University of Geosciences (Beijing), Beijing 100083, China. [2]School of Chemistry, Chemical Engineering and Biotechnology, Nanyang Technological University, Singapore 637459, Singapore. [3]Beijing Key Laboratory of Microstructure and Properties of Solids, Faculty of Materials and Manufacturing, Beijing University of Technology, Beijing 100124, China. [4]School of Physics, Beihang University, Beijing 100191, China. [5]School of Science, RMIT University, Melbourne, VIC 3000, Australia. [6]These authors contributed equally: Lizhen Liu, Jingcong Hu. ✉e-mail: chenfang@cugb.edu.cn; luyue@bjut.edu.cn; hhw@cugb.edu.cn

work showed that neighboring Fe SAs cleaved O-O bonds efficiently, which is a key step in the preferred four-electron oxygen reduction reactions in fuel cells[5]. In addition, the neighboring Pd SAs exhibited high activities and selectivities for cleavage of carbon–halogen bonds, which was difficult to achieve with either i-SAs or metallic Pd nanoparticles[6]. Although SACs have demonstrated fascinating catalytic activity in various catalytic reactions, regulation of the atomic configuration remains a challenge[7,8]. This is due in part to the tendency of neighboring SAs to form nanoclusters or nanoparticles, even under mild reaction conditions such as room temperature. Overcoming this challenge and achieving precise control of SAC configurations represents an important direction in the field of catalysis.

Noncentrosymmetric materials, particularly ferroelectrics, offer promising approaches to achieving switched surface properties and controlling surface atom loading configurations. However, the surface properties of ferroelectrics are susceptible to the effects of surface screening fields, which can lead to the adsorption of ions or charged particles and subsequent weakening of the polar nature and polarization electric field (PEF)[9,10]. To achieve a larger and more controllable polar nature and PEF, steerable domain switching is desirable. Our group has reported that the application of an external electric field regulated the polar natures and PEF of ferroelectrics, thus providing switched surface properties[11–14]. Domains can be switched along the direction of the external electric field when the intensity of the external field is higher than its coercive electric field, which causes drift of the positive and negative dipoles along opposite directions, leading to a maximum PEF[15,16]. The effective PEF provides a strong attraction for adsorption and anchoring of ions on the surface of the ferroelectric. Moreover, the PEF provides a driving force for charge separation in the bulk ferroelectric phase, allowing for simultaneous tuning of the surface and bulk driving forces. These characteristics offer ferroelectric photocatalysts unparalleled advantages for loading SACs.

Herein, we synthesized periodic one-dimensional (1D) SA arrays by utilizing $Bi_4Ti_3O_{12}$ (BTO) ferroelectric single-crystal nanosheets as the substrate and subsequently introducing corona poling treatment, which yielded stable 1D SA arrays with substantial neighboring atom sites. Corona poling switched the domain direction in the BTO while breaking the surface screening field, which endowed the domains in the well-polarized BTO (BTOP) with orderly charged properties to anchor Au SAs along the domains through electrostatic adsorption and played a crucial role in the syntheses of 1D SA arrays. Theoretical and experimental investigations were performed to reveal the crucial role and mechanism of this SA configuration in efficient $CO_2$ photoreduction. As a representative example, the 1D Au SA array structure provides neighboring Au sites to enhance $CO_2$ adsorption and activation via dual-site adsorption compared to single-site adsorption on Au i-SAs. Additionally, the 1D Au SA arrays anchored on the BTO surface prevented the domains from reversing and suppressing the depolarization of BTO, which gave rise to a stronger PEF and enabled fast transfer of photocharges from the bulk to surface catalytic sites with minimum loss. Thus, the optimized BTO sample with 1D Au SA arrays achieved a CO evolution rate of 34.15 $\mu mol \cdot g^{-1} \cdot h^{-1}$ without any photosensitizer or sacrificial reagents. Moreover, the proposed method was shown to be a general strategy by loading 1D Pt, Ag, Fe, Co, and Ni SA arrays on BTO nanosheets.

## Results

### Characterization of the as-prepared photocatalysts

BTO single-crystal nanosheets were synthesized by a hydrothermal method, which was followed by an ice-bath treatment to obtain $BTOAu_X$ (X = 1, 2, 3, 4) with different $HAuCl_4$ concentrations[17]. In addition, BTO was treated by corona poling to obtain BTOP, and then the BTOP was immediately treated by the above ice-bath treatment to prepare BTOPAu (Fig. 1a). BTO and BTOP deposited with the same loading amounts of Au nanoparticles (BTOAuNP and BTOPAuNP) were

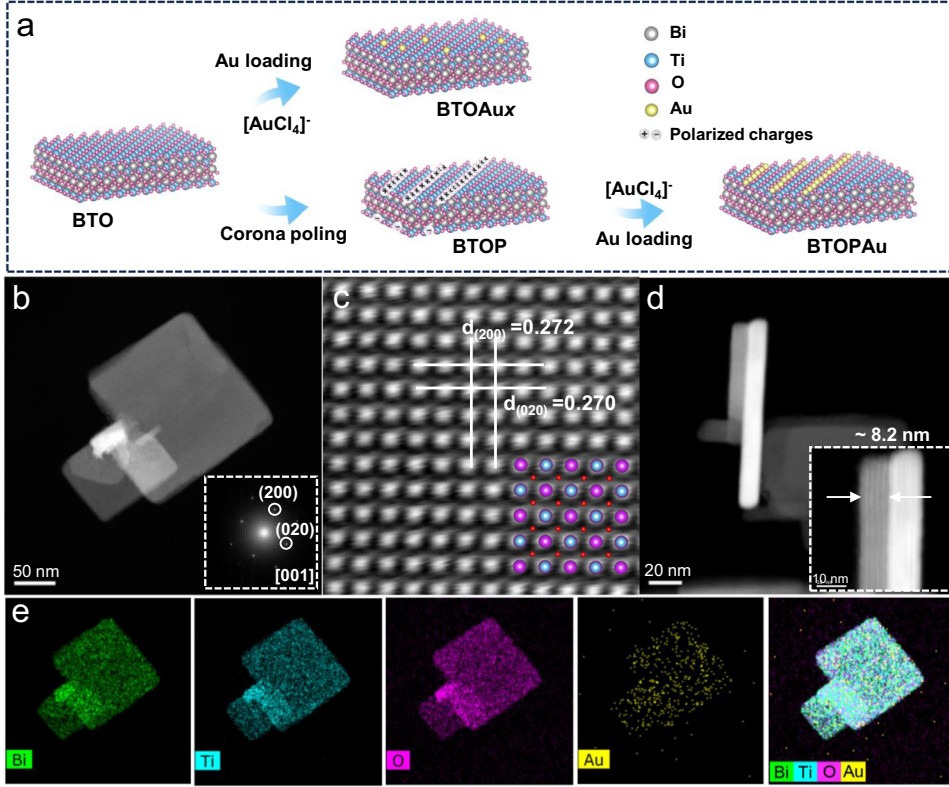

**Fig. 1 | Schematic showing the preparation process and characterization of the samples. a** Preparation process for BTO, BTOAu, and BTOPAu. **b, d** TEM images and SAED (insert in **b**), **c** HAADF-STEM image and **e** element maps for BTOPAu.

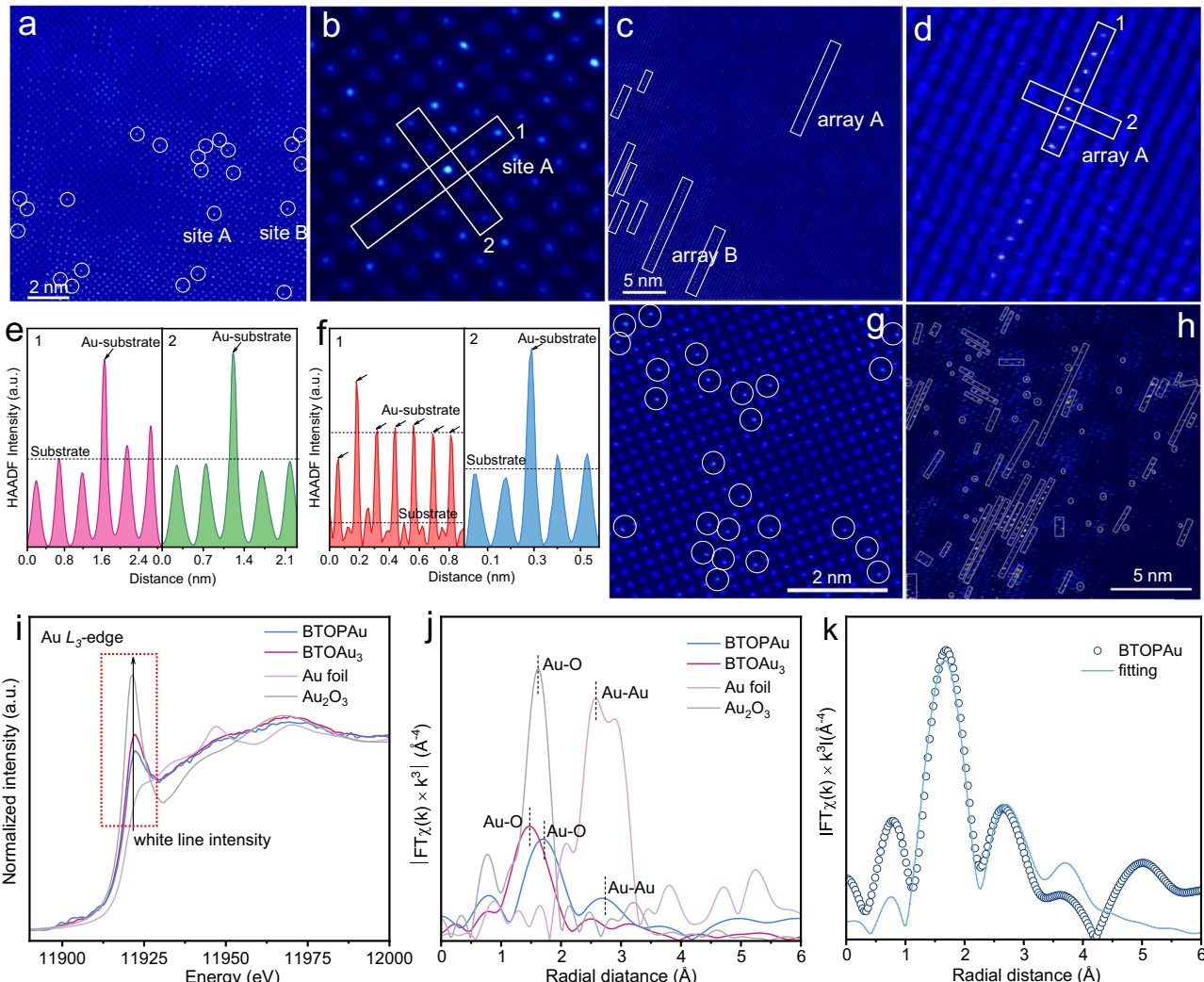

**Fig. 2 | Structural characterizations of the Au SAs on BTOAu₃ and BTOPAu.**
**a, b** HAADF-STEM images of BTOAu₃. **c, d** HAADF-STEM images of BTOPAu
(enlarged area from **a** and **b**). Line scan HAADF intensity profile of **e** site A (direction-1 and 2) and **f** array A (direction-1 and 2) versus distance. **g, h** Multiple HAADF-STEM images of (**b**) BTOAu₃ and (**d**) BTOPAu. **i** Au $L_3$-edge normalized XANES spectra and **j** Au $L_3$-edge EXAFS spectra in R space of BTOAu₃, BTOPAu, and references. **k** Fitting results of the $k^2$-weighted FT spectrum of BTOPAu at the Au $L_3$-edge.

also synthesized as reference samples by photodeposition. The X-ray diffraction (XRD) patterns suggested the preparation of orthorhombic BTO in all the samples. Corona poling and Au loading did not change the crystal structure and phase (Supplementary Fig. 1). The scanning electron microscopy (SEM) and transmission electron microscopy (TEM) images showed that all samples exhibited sheet-like morphologies with clean surfaces (Fig. 1b–d; Supplementary Fig. 2). Atomic force microscopy (AFM) revealed a range of BTOPAu thicknesses varying from ~6.2 nm to ~10.1 nm (Supplementary Fig. 3), indicating that the sample had a thin-layered structure. However, the TEM images of BTOAuNPs displayed observable Au nanoparticles loaded on the surface of the BTO (Supplementary Fig. 4). The selected area electron diffraction (SAED) pattern showed that BTOPAu had a single-crystal structure, and the observed lattice spacings of 0.270 and 0.272 nm corresponded to the (020) and (200) crystal planes of BTOPAu[18]. Elemental maps of Bi, Ti, O and Au revealed a uniform spatial distribution of Au on the surface of BTO (Fig. 1e). The inductively coupled plasma (ICP) results revealed Au loadings of 0.11 wt%, 0.15 wt%, 0.27 wt%, 0.44 wt%, and 0.30 wt% on BTOAu₁, BTOAu₂, BTOAu₃, BTOAu₄, and BTOPAu, respectively (Supplementary Table 1). BET results indicated that the specific surface areas of BTOAu₃ and BTOPAu were 11.74 m²/g and 12.25 m²/g, respectively. The Au coverage was determined

to assess the performance difference of the photocatalysts with the following formula:

$$C_{Au} = \frac{N_{Au} \times S_{Au}}{S} \qquad (1)$$

where $C_{Au}$ is the Au atom coverage, $N_{Au}$ is the number of Au atoms, $S_{Au}$ is the cross-sectional area of Au and $S$ is the surface area of the substrate. Consequently, $C_{Au}$ was determined for BTOAu₃ and BTOPAu to be ~6.13% and 6.51%, respectively. This demonstrated that BTOAu₃ and BTOPAu had comparable Au atom coverage.

To disclose the distribution of Au SAs on the BTO surface, atomic resolution high-angle annular dark-field scanning TEM (HAADF-STEM) was conducted. The bright dots on the surfaces of BTOAu₃ and BTO-PAu (Fig. 2a, c) were unambiguously ascribed to Au SAs according to the Z-contrast difference between the Au atom and the substrate Bi and Ti atoms[19]. Remarkably, there were two different permutations of Au SAs on the surfaces of BTOAu₃ and BTOPAu. On the surface of BTOAu₃, scattered bright dots were observed, indicating the formation of Au i-SAs (Fig. 2a, b). In sharp contrast, the bright dots were arranged to form an array structure on BTOPAu (Fig. 2c, d). The corresponding quantitative intensity analyses of the bright dots confirmed the

discrete permutations of the Au SAs on BTOAu$_3$. The two-dimensional intensity analyses for site A and site B (Fig. 2b, e; Supplementary Fig. 5a–c) indicated that the HAADF intensities of the bright dots were higher than those of the surrounding atoms, demonstrating the formation of Au i-SAs. For BTOPAu, array A and array B displayed two intensity levels along direction-1 (Fig. 2d; Supplementary Fig. 5d), and the low-intensity level corresponded to Bi/Ti atoms and the high level indicated periodic Au atom loading (Fig. 2f; Supplementary Fig. 5e). The HAADF intensity curves also showed only one high-intensity dot along direction 2 (Fig. 2f; Supplementary Fig. 5f), which confirmed the 1D Au SA array structure. The linear arrays of Au SAs were attributed to the corona poling treatment, which switched the domain direction of BTO, provided orderly charges and formed periodic permutations of the positive and negative dipoles. Then, the Au precursor was absorbed on the positive dipole sites to form 1D arrays due to the Coulomb force. In addition, a quantitative analysis of the distribution of Au atoms via statistical analyses of the HAADF-STEM images illustrated that $C_{Au}$ was ~6.24% and 6.65% for BTOAu$_3$ and BTOPAu, respectively, confirming the above ICP results (Fig. 2g, h).

To disclose the Au SA coordination environment, X-ray absorption near-edge structure (XANES) and extended X-ray absorption fine structure (EXAFS) spectroscopy were conducted. The white line intensity of the XANES analysis indicated that the stable Au oxidation state excluded the formation of Au clusters or nanoparticles, and the peak intensities of BTOAu$_3$ and BTOPAu were between that of Au foil and the Au$_2$O$_3$ reference (Fig. 2i)[20,21]. The EXAFS data and fitting analyses indicate the coordination environments of the Au SAs on BTOAu$_3$ and BTOPAu. The Fourier transformed EXAFS (FT-EXAFS) spectra and fitting parameters indicated that the Au SAs were coordinated with O atoms on the surfaces of BTOAu$_3$ and BTOPAu (Fig. 2j; Supplementary Fig. 6 and Table 2). The distinct Au-O coordination was in stark contrast with the metallic Au-Au coordination observed in Au foil. A small peak at ~2.90 Å was observed for BTOPAu, corresponding to the second-shell Au-Au coordination. This was proven by the fitting analysis, which confirmed the formation of a Au-O-Au structure in the Au SA array structure (Fig. 2k). Higher Au $4f_{7/2}$ and Au $4f_{5/2}$ binding energies for BTOAu$_3$ ($4f_{7/2}$: 84.2 eV, $4f_{5/2}$: 88.2 eV) and BTOPAu ($4f_{7/2}$: 84.1 eV, $4f_{5/2}$: 88.0 eV) compared to those for metallic Au ($4f_{7/2}$: 83.9 eV, $4f_{5/2}$: 87.9 eV) confirmed the presence of oxidized Au SAs in BTOAu$_3$ and BTOPAu (Supplementary Fig. 7a)[22,23]. In addition, the O atom binding energies of BTOAu$_3$ and BTOPAu showed shifts compared to those in BTO and BTOP, which verified their distinct local atomic permutations and the coordination of Au and O atoms (Supplementary Fig. 7b). The shifts of two peaks at 281.6 cm$^{-1}$ to 277.2 cm$^{-1}$ and 326.8 cm$^{-1}$ to 334.3 cm$^{-1}$ in the Raman spectra corresponded to the vibrational bands in [TiO$_6$] octahedra, which indicated that the Au-O bonds influenced the vibrations of the [TiO$_6$] octahedra (Supplementary Fig. 8)[24]. The above characterization results clearly indicated that Au SAs were successfully anchored on the BTO substrates and formed different configurations on BTOAu$_3$ (i-SAs) and BTOPAu (1D SA arrays).

To explore the universality of the current PEF strategy, BTO and BTOP with various metal atoms, including Pt, Fe, Co, Ni, and Ag, were prepared. The corresponding HAADF-STEM images are shown in Supplementary Figs. 9 and 10. The metals loaded on BTO and BTOP showed remarkably different permutations, and the Pt, Fe, Co, Ni, and Ag metal atoms were dispersed and anchored on the surface of BTO to form i-SAs. However, similar to Au atoms, all of the above metal atoms formed SA arrays linearly assembled on BTOP. This demonstrated that the polarization field strategy can be applied generally to prepare 1D SA arrays.

## Photocatalytic CO$_2$ reduction

The CO$_2$ photoreduction capabilities of the as-obtained samples were examined in a gas–solid system and a liquid–solid system under simulated solar light without any sacrificial agents[25]. In the gas–solid system, pristine BTO yielded CO as the main product with an evolution rate of 1.92 μmol·g$^{-1}$·h$^{-1}$, and it increased to 6.41 μmol·g$^{-1}$·h$^{-1}$ for BTOP, revealing the advantage of the PEF in carrier separation (Fig. 3a, c). The incorporation of Au i-SAs substantially improved the CO$_2$ reduction rate of BTO, with an optimal CO yield of 27.55 μmol·g$^{-1}$·h$^{-1}$ achieved for BTOAu$_3$ among the BTOAu$_X$ ($X$ = 1, 2, 3, 4) series. Notably, all of the BTOAu$_X$ catalysts showed better performance than BTOAuNP, indicating the superiority of the SACs in improving photocatalytic activities (Fig. 3a). Compared with the Au i-SAs, the 1D Au SA arrays showed substantially improved CO$_2$ photoreduction, and the CO formation rate of BTOPAu reached 34.15 μmol·g$^{-1}$·h$^{-1}$; these were ~4 times and 18 times higher than those of BTOAuNP and pristine BTO, respectively, revealing the catalytic advantages of the special 1D array configuration (Fig. 3a, c). The production rate was superior to those of most photocatalysts under similar reaction conditions (Supplementary Table 3). To evaluate whether polarization enhanced the performance of the metal nanoparticles, we synthesized BTOPAuNP and determined a CO yield of 34.10 μmol·g$^{-1}$ within 4 h, a value similar to that of BTOAuNP (34.86 μmol·g$^{-1}$) (Supplementary Fig. 11). Therefore, it was concluded that the PEF did not obviously enhance the photocatalytic CO$_2$ reduction rate of BTOPAuNP. The improved performance of the Au SAs was attributed to the PEF altering the permutations of i-SAs to periodic Au SA arrays with neighboring Au catalytic sites. The photoreduction products also contained trace amounts of H$_2$ and CH$_4$, which indicated the unfavorable H$_2$ evolution competing reaction in the gas–solid system and harsh eight-electron pathway (Fig. 3c). After a 12 h photocatalytic reaction, BTOPAu still maintained high photocatalytic activity and stability (Supplementary Fig. 12). In addition, BTOPAu also showed the best CO$_2$ photoreduction to CO among the selected samples in the liquid–solid system, reaching 57.77 μmol·g$^{-1}$ after 4 h (Fig. 3b). Control experiments indicated that no CO was detected in the absence of a CO$_2$ atmosphere or under illumination, and a negligible amount of CO was generated without the catalysts due to CO$_2$ splitting under the simulated solar light (Fig. 3d). A $^{13}$CO$_2$ isotopic labeling experiment showed a peak at $m/z$ = 29 ($^{13}$CO) in the mass spectrum, clarifying the origin of CO (Fig. 3d).

In situ Fourier transform infrared spectroscopy (FTIR) was conducted to analyze the interactions of CO$_2$ and BTOPAu (Fig. 3e). There were three peaks at 1700 cm$^{-1}$, 1623 cm$^{-1}$, and 1541 cm$^{-1}$, which were assigned to two intermediates (*CO$_2^-$ and *COOH) in the CO$_2$ reduction process and confirmed CO$_2$ activation under illumination[26]. A peak at 1421 cm$^{-1}$ corresponded to the symmetric stretching vibration of *HCO$_3^-$, which was also a vital signal of CO$_2$ activation. Importantly, a peak at 2019 cm$^{-1}$ for *CO continuously increased with illumination time, which suggested steady formation of CO during the reaction process and the conversion of CO$_2$ to CO. Based on the above results and previous work, a possible CO$_2$ photoreduction mechanism is proposed:

$$CO_2 \longrightarrow {}^*CO_2; H_2O \longrightarrow H^+ + OH^- \tag{2}$$

$$^*CO + e^- \longrightarrow {}^*CO_2^- \tag{3}$$

$$^*CO_2^- + H^+ \longrightarrow {}^*COOH \tag{4}$$

$$^*COOH + H^+ + e^- \longrightarrow {}^*CO + H_2O \tag{5}$$

$$^*CO \longrightarrow CO \tag{6}$$

where "*" represents adsorption and e stands for electrons.

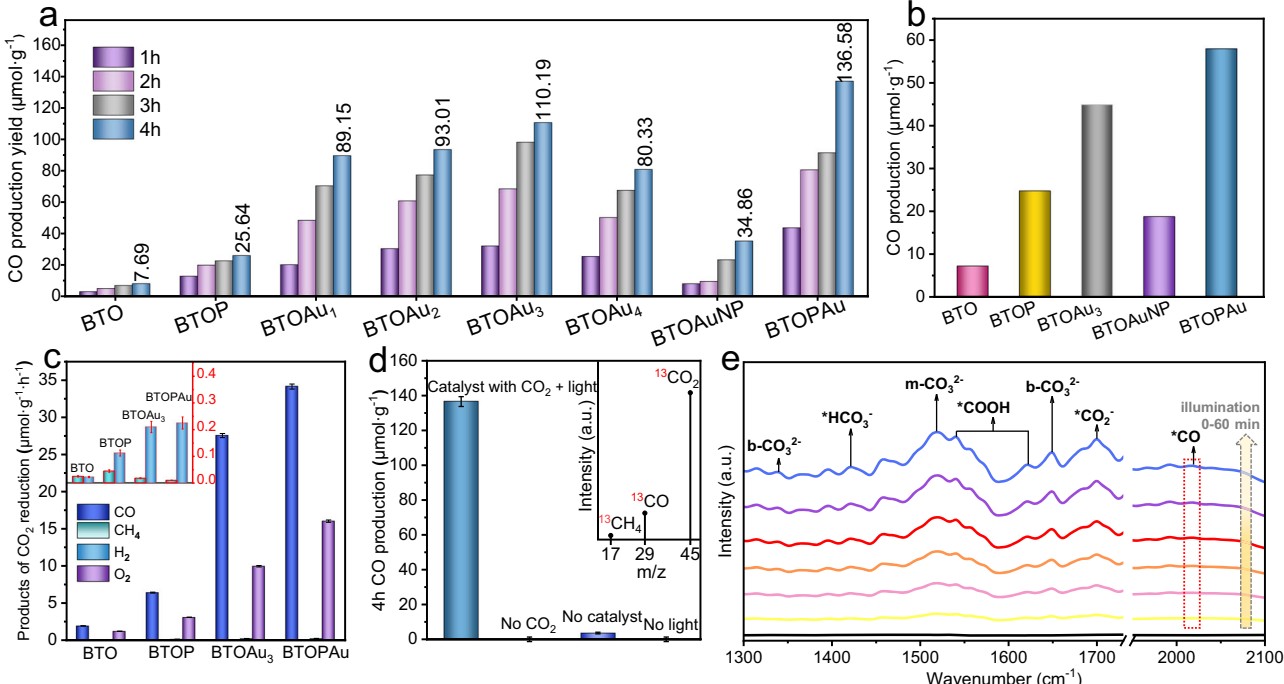

**Fig. 3 | CO₂ photoreduction and in situ FTIR spectroscopy. a** CO production from CO₂ photoreduction over BTO, BTOP, BTOAuₓ (X = 1, 2, 3, 4), BTOAuNP and BTOPAu samples in a gas–solid system under simulated solar light. **b** Photoreduction of CO₂ to CO over BTO, BTOP, BTOAu₃, BTOAuNP and BTOPAu samples for 4 h in a gas–liquid-solid system under simulated solar light. **c** Evolution rates for CO, CH₄, H₂ and O₂ over BTO, BTOP, BTOAu₃ and BTOPAu. **d** Corresponding production of CO over BTOPAu under different conditions for 4 h (inset of (**d**): mass spectra of ¹³CO originating from ¹³CO₂ photoreduction over BTOPAu). **e** In situ FTIR spectra for CO₂ photoreduction over BTOPAu under a 300 W Xe lamp.

## Mechanism of the SAC configuration-dependent catalytic performance

Based on the photocatalytic process, including light absorption, charge transfer and surface catalysis, the effects of the SAC configuration on these crucial steps were analyzed to elucidate the origin of the distinct CO₂ photoreduction activity. BTO, BTOP, BTOAu₃, and BTOPAu have similar band structures, all of which meet the thermodynamic requirements for CO₂ reduction into CO (Supplementary Figs. 13 and 14).

The separation and transport of photogenerated charges were studied with photoelectrochemical measurements. The linear sweep voltammetry (LSV) curves and transient photocurrent responses show that BTOPAu exhibited the highest cathode current density, demonstrating that the photogenerated charge carriers were separated by the synergy of the 1D Au SA arrays and the enhanced PEF (Fig. 4a). Electrochemical impedance spectroscopy (EIS) showed that the semicircle radius for BTOPAu was significantly smaller than those of the other three photocathodes, indicating that BTOPAu had the weakest resistance to electron flow and the fastest interfacial charge transfer (Fig. 4b). The average lifetime was increased from 3.87 ns (BTO) to 4.06 ns (BTOPAu), as revealed by time-resolved fluorescence emission decay spectra and quenching of the photoluminescence (PL) signal from BTOPAu, provided further evidence for suppressed recombination of the photogenerated carriers (Fig. 4c; Supplementary Fig. 15). The high separation efficiency mainly originated from efficient charge separation in bulk and on the surface of BTOPAu, in which effective surface charge separation was attributed to the surface Au SA loading and the bulk charge separation was due to the migration driving force provided by PEF. To obtain deeper insight, the electric hysteresis loops were surveyed. BTOAu₃ and BTOPAu showed larger remnant polarizations (Pr) than BTO, which was attributed to the Au SAs anchoring the domains and keeping them in an aligned polarization state (Fig. 4d)[27]. This fascinating phenomenon was

attributed to Au SA arrays anchored by the PEF, which played a pinning role by inhibiting reversion of the domains into disordered states after withdrawing the applied voltage; this maintained a strong PEF for persistent separation of the photogenerated charges. This was also observed for oxygen vacancy-decorated ferroelectrics[11]. Notably, the increased maximum polarization in both the BTOP and BTOPAu samples compared to BTO and BTOAu₃ was a direct result of corona poling switching the domains within the samples, leading to uniform arrangements of the domains. Additionally, the Pr of BTOPAu was slightly higher than that of BTOAu₃, which was attributed to the fact that different Au configurations exhibited varying degrees of this effect. Then, the fast charge dynamics were analyzed. Na₂SO₃ used to simulate hole scavenging can provide a charge separation efficiency of 100%, and the electrons all fall into the current loop. In the current situation, the surface charge transfer efficiency was 67% for BTOPAu, far higher than that of BTO (15%), which confirmed the accelerated charge dynamics in BTOPAu (Supplementary Fig. 16)[28]. The surviving carriers arriving near the surface directly participated in CO₂ activation and reduction. The surface potentials of samples were determined with scanning Kelvin probe force microscopy (SKPFM) to verify these results (Fig. 4e, f). For comparison, the potential is displayed as the absolute value of the surface potential (Δ_ave). Before illumination, BTOP, BTOAu₃, and BTOPAu all showed larger surface potentials than BTO, indicating that the presence of the PEF and Au SAs enhanced the surface electric field of BTO. After illumination, all samples exhibited higher surface potentials compared to those in the dark. This indicated increased charge separation and redistribution during photoexcitation of all synthesized samples. The difference in the surface potential (ΔΦ) before and after illumination is defined as follows:

$$\Delta\Phi = \Delta_{ave(illumination)} - \Delta_{ave(dark)} \qquad (7)$$

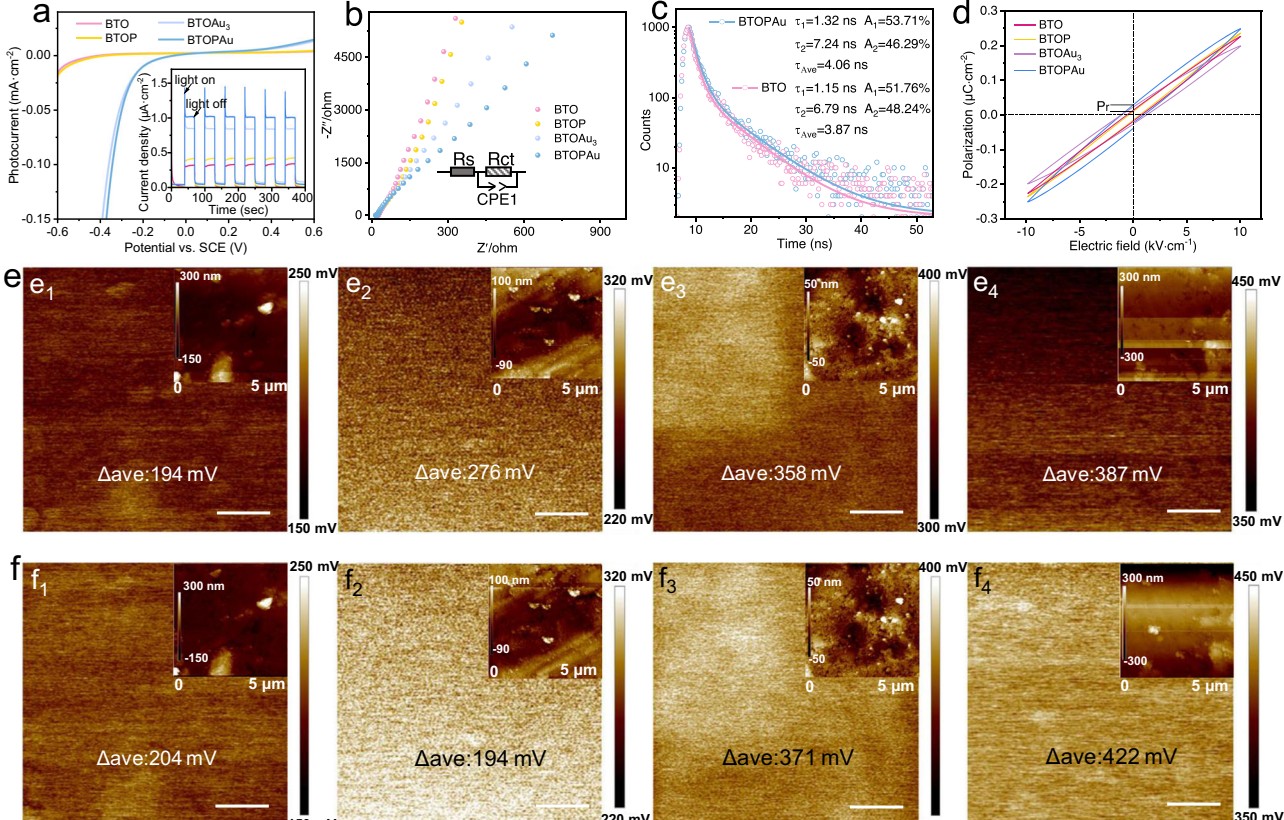

**Fig. 4 | Charge transfer and surface potential distribution. a** LSV curves (inset: transient photocurrent response; applied voltage 0 V) and **b** EIS Nyquist plots with the fitted equivalent circuit for BTO, BTOP, BTOAu₃ and BTOPAu. **c** Fluorescence emission decay spectra of BTO and BTOPAu. **d** Electric hysteresis loops of BTO, BTOP, BTOAu₃, and BTOPAu (at 10 kV·cm⁻¹). **e, f** Surface potential distributions corresponding to BTO, BTOP, BTOAu₃, BTOPAu **e₁–e₄** before and **f₁–f₄** after illumination (inset: AFM images; Δ_ave is the absolute value of the surface average potential; scale bar: 1 μm).

Here, $\Delta\Phi$ was determined to be 10 mV for BTO, 25 mV for BTOP, 13 mV for BTOAu₃, and 35 mV for BTOPAu. This confirmed that both PEF and Au SA loading played important roles in promoting charge separation. The largest $\Delta\Phi$ observed in BTOPAu indicated the formation of the most robust electric field, which was attributed to the synergistic effect of the PEF and Au SA loading, thus largely facilitating rapid separation and transfer of the photogenerated charges and contributing to the enhanced photocatalytic activity of BTOPAu. Moreover, $CO_2$ adsorption reflected the affinity between the photocatalyst and $CO_2$. BTOPAu showed the highest $CO_2$ adsorption capacity, followed by BTOAu₃, which were much higher than those BTO and BTOP, confirming that the Au SAs served as absorption sites for $CO_2$ on BTO (Supplementary Table 4). It also indicated that the 1D SA array configuration was more advantageous for $CO_2$ adsorption than the i-SAs, given the similar actual Au loading concentrations on BTOPAu and BTOAu₃. The neighboring Au atoms may have provided dual adsorption sites for more efficient adsorption of $CO_2$, which was supported by the density functional theory (DFT) calculations shown below.

Theoretical simulations demonstrated that the Au i-SAs spontaneously formed a stable structure on the surface of BTO, while loading of three neighboring Au SAs (imitated 1D Au SA array) on BTO was thermodynamically unfavorable compared to Au i-SAs (Fig. 5a). In contrast, the formation of a 1D Au SA array was spontaneous and stable when a voltage was applied. This explains why various 1D SA arrays were formed on BTOP, as shown by the HAADF-STEM results. To provide atomic-level insight into the crucial roles of the SA configuration in the surface catalytic reactions, the charge transfer behavior and $CO_2$ absorption of the Au i-SAs and neighboring Au SAs were analyzed (Fig. 5b, c; Supplementary Figs. 17 and 18). The results

indicated that Au SA sites were charge accumulation and transfer sites, which were the main catalytic sites for adsorbing and activating $CO_2$. The simulated charge distribution profile revealed that the Au atom exhibited a charge interaction with BTO, as evidenced by the 3D charge density difference, where charge transfer around the surrounding Au atom was clearly observed (Supplementary Fig. 19). Moreover, $CO_2$ was adsorbed on the Au i-SAs through Au-O=C=O single-site adsorption (Fig. 5d–f). Notably, dual-site adsorption as Au-O=C=O-Au occurred preferentially on the neighboring Au SAs, which afforded more intense electron transfer and stronger affinity between the Au and $CO_2$. The Gibbs free energies for $CO_2$ reduction over BTO, BTOAu₃ (BTO with Au i-SAs) and BTOPAu (BTO with neighboring 1D Au SAs) were also calculated based on the results of in situ FTIR spectroscopy (Fig. 5g; Supplementary Figs. 20 and 21). Formation of the adsorbed intermediate *COOH was found to be the potential limiting step in $CO_2$ reduction. $CO_2$ adsorption and the formation of *COOH over all samples were endothermic processes, while the formation of *CO and the desorption of CO were exothermic. Remarkably, BTOPAu demonstrated lower reaction barriers for the endothermic processes compared to BTOAu₃ and BTO, which indicated easier $CO_2$ activation and conversion on BTOPAu for more efficient photocatalytic activity, confirming the experimental results.

## Discussion

In this work, we propose that the PEF-induced periodic 1D Au SA arrays on ferroelectric BTO single-crystal nanosheets through corona poling followed by an ice-bath treatment. The as-prepared BTOP assembled with 1D SA arrays exhibited prominent $CO_2$ photoreduction activity without any photosensitizer or sacrificial agent. Experimental and

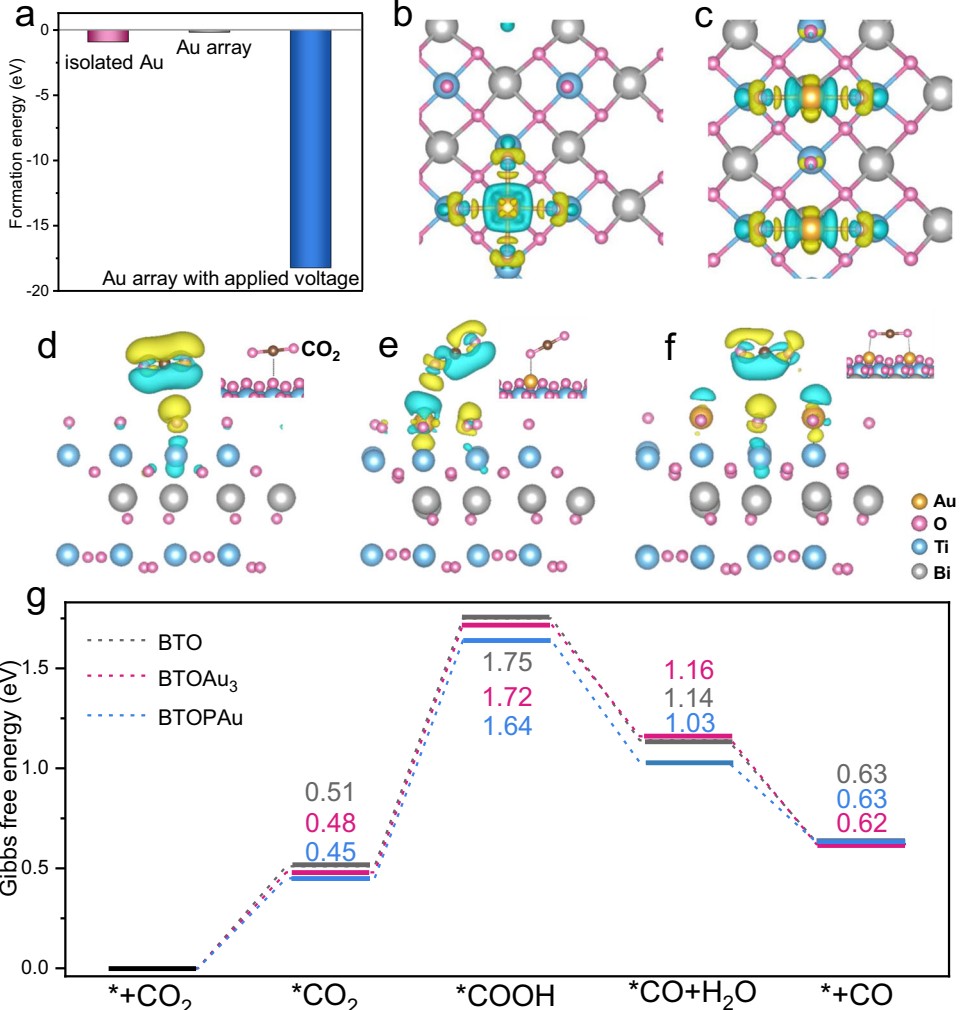

**Fig. 5 | Theoretical study. a** Theoretically calculated formation energies of isolated Au atoms on BTO and Au arrays on BTO without and with an applied voltage. Charge difference of **b** BTOAu₃ and **c** BTOPAu. Charge difference of $CO_2$ adsorbed on **d** BTO, **e** BTOAu₃ and **f** BTOPAu (charge accumulation is shown in yellow and depletion in blue). **g** Gibbs free energy diagrams for $CO_2$ photoreduction to CO over BTO, BTOAu₃ and BTOPAu.

theoretical results showed that corona poling switched the directions of domains to regular PEFs for 1D SA array anchoring. The 1D Au SA arrays had a pinning effect and inhibited reversion of the domains into disordered states after withdrawing the applied voltage, which maintained a strong PEF driving persistent separation and transport of the photogenerated charges. The high photocatalytic performance also originated from the fact that the 1D Au SA arrays showed great dispersion with copious neighboring catalytic sites, which allowed dual-site adsorption to lower the energy barriers for the $CO_2$ conversion process compared to those of the Au i-SAs. This general strategy was also confirmed by loading uniform 1D SA arrays of various metals, e.g., Pt, Fe, Co, Ni, and Ag, on the surface of BTOP. This may enable the syntheses and regulation of SCAs with special configurations.

## Methods

### Synthesis of BTO

The synthetic method was based on a previously reported approach[25]. Bi(NO₃)₃·5H₂O (0.80 g), 3.60 g of NaOH, and 0.136 mL of TiCl₄ were dissolved in 30 mL of deionized water under ultrasonic treatment for 30 min and then stirred for 1 h. The above slurry was added to a 50 mL Teflon-lined stainless steel autoclave for hydrothermal treatment at 220 °C for 12 h, after which the autoclave was cooled to room temperature. The samples were washed with deionized water and ethanol

three times. Then, 0.364 g BTO samples were obtained for the following experiments after drying at 60 °C for 12 h.

### Synthesis of BTOAu_X (X = 1, 2, 3, 4)

The BTOAu_X (X = 1, 2, 3, 4) samples with different Au loadings were synthesized by treating BTO in HAuCl₄ solutions. First, 0.10 g of BTO and 0.02 g of trisodium citrate were dissolved in 20 mL of deionized water and stirred (Solution A). A series of 1 g/100 mL HAuCl₄ solutions (0.05, 0.10, 0.20, and 0.40 mL) were added to 10 mL of deionized water and then frozen with liquid nitrogen. The ice was transferred into Solution A and stirred for 1 h. Then, the samples were washed with deionized water and finally dried at −50 °C for 12 h. The samples with HAuCl₄ solutions of 0.05, 0.10, 0.20, and 0.40 mL in the reaction solution were denoted as BTOAu₁, BTOAu₂, BTOAu₃, and BTOAu₄, respectively. This treatment was inspired by the reported approach[25].

### Synthesis of BTOP

BTOP was obtained by polarization treatment of BTO, which was executed with a laboratory-made corona-poling instrument based on a previous report[12,29]. Typically, 0.20 g of BTO powder was put into the sample holder, and then a 5 kV DC voltage was applied between the mold and a metallic needle. The polarized electric field was calculated to be 25 kV/cm, and the samples were poled for 30 min. Afterward,

0.192 g of a powder was collected carefully, which was denoted as BTOP.

## Synthesis of BTOPAu

A 0.10 g BTOP sample was immediately collected and put into 20 mL of deionized water with 0.02 g of trisodium citrate. The ice containing 0.02 mL of the $HAuCl_4$ solution and 10 mL of deionized water was transferred into the BTOP solution and stirred for 1 h. Then, the sample was washed with deionized water and finally dried at −50 °C for 12 h to obtain BTOPAu.

## Syntheses of BTOAuNP and BTOPAuNP

First, 0.10 g of the BTO sample powder and 0.20 mL of a 1 g/100 mL $HAuCl_4$ solution were mixed in 30 mL of deionized water. To achieve adsorption-desorption equilibrium, the suspension was stirred in darkness for 30 min and then irradiated by a 300 W mercury lamp with continuous stirring. Then, the sample was washed with deionized water and dried at −50 °C for 12 h to obtain BTOAuNP. By employing an analogous method, the substitution of BTO with BTOP was undertaken to synthesize BTOPAuNP.

## Catalyst characterization

Powder XRD was performed with a D8 Advance diffractometer (Bruker, Germany) with Cu Kα radiation ($\lambda$ = 0.15418 nm) over the $2\theta$ range 5°–80°. Morphologies and microstructures were observed by scanning electron microscopy (SEM, Hitachi S-4800, Japan). For the syntheses of the SEM samples, the powder was first dispersed in 10 mL of absolute ethanol followed by ultrasonication for 20 min, and then a small amount of liquid was dropped onto a silicon pellet and dried in air for 20 h. The atom-level microstructure was observed by scanning transmission electron microscopy (STEM), and Cs-corrected STEM images were obtained on an FEI Titan G2 60–300 kV microscope operating at 300 kV. These microscopes were equipped with probe spherical aberration correctors, enabling sub-angstrom imaging with HAADF-STEM detectors. For HAADF-STEM imaging, the inner and outer collection angles were 58.5 and 200 mrad, respectively. The UV–vis diffuse reflectance spectra (DRS) were measured with a Varian Cary 5000 UV–vis spectrophotometer with pure white $BaSO_4$ as the reference. X-ray photoelectron spectroscopy (XPS) was conducted with an ESCALAB 250Xi spectroscopy instrument (Thermo Fisher, Britain) to determine the surface states and chemical composition. The surface potentials were characterized by Kelvin probe force microscopy (KPFM) (Bruker icon, Germany). The ferroelectric properties were measured with a ferroelectric tester (aixACCT Systems GmbH, Aachen, Germany). In situ Fourier transform infrared (FTIR) spectroscopy was implemented with a Nicolet IS50 instrument with an in situ Harrick IR cell under a 300 W Xe lamp. Photoluminescence emission (PL) spectra were obtained with a fluorescence system (Hitachi, F-4600, Japan) with a 150 W xenon lamp as the excitation source. Time-resolved fluorescence decay spectra were collected on a fluorescence spectrophotometer (FLSP-920, Edinburgh Instruments). $CO_2$ adsorption curves were obtained with a Micromeritics instrument (ASAP 2020). The characterizations are carried out under relevant guidance[29].

## Photoelectrochemical measurements

The transient photocurrents, Mott–Schottky plots, electrochemical impedance spectra, and linear sweep voltammetry were obtained with an electrochemical workstation (CHI660E, Chenhua Instrument Co. Shanghai, China) furnished with a standard three-electrode system with a saturated calomel electrode (SCE) and platinum wire as the reference electrode and counter electrode, respectively. The test methods were based on previous reports[29]. The as-synthesized samples coated onto the indium-tin oxide (ITO) sheet glass (20 mm × 45 mm × 1.1 mm) were used as the working electrode. Ten milligrams of the powder was dispersed in 1 mL of absolute ethanol followed by ultrasonication for 30 min, and then the suspension was evenly dripped onto the ITO sheet glass substrate and dried at room temperature for 24 h. An aqueous solution of 0.10 M $Na_2SO_4$ was used as the electrolyte, and a 300 W xenon lamp was employed as the light source. A 0.10 M $Na_2SO_3$ solution served as a hole scavenger. The applied voltage of the photocurrent response was 0.0 V.

## Photocatalytic CO$_2$ reduction

The photocatalytic $CO_2$ reductions were investigated in a gas–solid and gas–liquid-solid reactor with a 300 W Xe lamp (Perfectlight, China) as the light source. For the gas–solid reaction, 1.70 g of $NaHCO_3$ was put into the bottom of the reactor cell and subjected to an exhaustive vacuum treatment. Then, 15 mL of $H_2SO_4$ was injected into the cell to react with $NaHCO_3$ and generate $CO_2$ gas (1 atm). After the photoreduction reaction, 1 mL of the resultant gas was qualitatively analyzed with a GC-2014C gas chromatograph (Shimadzu, Japan) equipped with an FID detector (carrier gas, Ar; FID-R temperature, 150 °C; FID-L temperature, 150 °C; chromatographic column temperature, 60 °C) and by mass spectrometry with a quadrupole-type mass spectrometer (OmniStar 300, carrier gas: He). $^{13}CO_2$ (Beijing Gaisi Chemical Gases Company) was used for the detection of the carbon sources. Gas–liquid-solid $CO_2$ photoreductions were carried out in the same reactor with 20 mL of deionized water and pure $CO_2$ gas. All measurements were carried out at room temperature.

## DFT calculation

The structure relaxation and single-point energies were calculated by density functional theory (DFT) calculations through DFT + U with $U_{eff}$ = 4.2 eV, which were implemented with the Vienna Ab initio Simulation Package (VASP) and the generalized gradient approximation (GGA) of Perdew−Burke−Ernzerhof (PBE). The related computational parameters (the plane wave cutoff energy of 400 eV, the k-point with a density of $(3 \times 3 \times 1)$ points in the Brillouin zone of the unit cell and the threshold of the self-consistent-field energy convergence of $10^{-4}$) were set for high-precision calculations. All of the computational parameters were set as fine to confirm the accuracy of the present results. A vacuum region of 15 Å in the $c$ direction of the $Bi_4Ti_3O_{12}$ crystal structure was created, and an applied electric field of $9 \times 10^{-24}$ V/Å was used in the calculation of the formation energy.

The formation energy ($E_f$) was calculated with the following formula:

$$E_f = E_{tot} - E_{sub} - n\mu_{Au} \tag{8}$$

where $E_{tot}$ and $E_{sub}$ are the total energy of the substrate structure with Au loading and the substrate structure, respectively, $n$ is the number of Au atoms, and $\mu_{Au}$ is the energy of one Au atom.

The adsorption energy ($E_{ads}$) was calculated with the following formula[25]:

$$E_{ads} = E_{tot} - E_{sub} - E_{co_2} \tag{9}$$

where $E_{tot}$, $E_{sub}$ and $E_{co_2}$ are the total energy of the adsorption structure, the substrate structure, and the $CO_2$ molecule, respectively.

The structures from the theoretical simulations of $BTOAu_3$ and BTOPAu were modeled with BTO with a single Au atom and BTO with dual Au atoms, respectively.

## Data availability

Full data supporting the findings of this study are available within the article and its Supplementary Information. The source data generated in this study are available in the figshare repository (https://doi.org/10.6084/m9.figshare.24558553)[30]. Additional data are available from the corresponding authors upon reasonable request. Source data are provided with this paper.

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

## Acknowledgements

H. H. acknowledges funding support from the National Key Research and Development Program of China (No. 2022YFB3803600), the National Natural Science Foundation of China (No. 52272244, 51972288, 12274009, and 12074016), and the Fundamental Research Funds for the Central Universities (2652022202). Y. L. acknowledges the support of the Beijing Innovation Team Building Program (Grant No. IDHT20190503), the Beijing Natural Science Foundation (Z210016), the Research and Development Project from the Shanxi-Zheda Institute of Advanced Materials and Chemical Engineering (2022SX-TD001) and the General Program of Science and Technology Development Project of Beijing Municipal Education Commission (KM202110005003). L. L. appreciates the China Scholarship Council.

## Author contributions

L. L. performed most of the experiments and theoretical simulation and wrote the first version of the paper; J. H. carried out the HAADF-STEM analysis; Z. M. performed the in situ FTIR analysis; Z. Z. helped to perform the theoretical simulation; B. H. helped to perform the electric hysteresis loops; R. X., Y. Z., T. M., and M. S. helped to revise the paper; H. H., Y. L., and F. C. initiated the research and revised the paper. All the authors discussed the results and commented on the paper.

## Competing interests

The authors declare no competing interests.
