## [Peer Review File · Nature Communications]

One-dimensional single atom arrays on ferroelectric nanosheets for enhanced CO₂ photoreductionREVIEWER COMMENTS

Reviewer #1 (Remarks to the Author):

The manuscript propose that utilizing PEF to induce the synthesis of periodic 1D Au SAs arrays on ferroelectric Bi₄Ti₃O₁₂ single-crystal nanosheets through Corona poling followed by ice-bath treatment. The as-prepared poled BTO assembled with 1D SAs arrays exhibits prominent CO₂ photoreduction activity. However, some critical issues should be addressed.

1. The XRD patterns of the reference material (like BTOAuNP) should be given.
2. The authors claimed that the all samples exhibit a thickness of 5 unit-cell layers. The authors should supplement the related experiments (like AFM) to further study thickness of the samples.
3. Figure 2d only shows a Au SAs array. Is this random? What is the probability of forming an array? Authors should provide a wider view of the HADDF-STEM images to show that the array formation is not random.
4. The signal noise ratio is low in Figure 2c, resulting in the possibility of getting untrusted data. Thus, authors should repeat experiments.
5. Figure 2j shows a small peak around 2.90 Å in BTOPAu, corresponding to the second shell Au-Au coordination. But the authors claimed that the the formation of Au-O-Au structure in Au SAs array structure. Why not the Au-O-Bi or Au-O-Ti structure or others? Please add sufficient tests to prove this. Besides, the detailed fitting curves and parameters should be given.
6. The authors claimed that the Pt, Fe, Co, Ni and Ag metal atoms are dispersedly anchored on the BTO surface to form i-SAs. However, the Supplementary Fig. 7 can not distinguish between different metal i-SAs. Please add the related experiments (like EDS) to verify the different metal elements on the BTO surface.
7. As we all know, in the presence of Au, light will induce the surface plasmon field, so is the dominant factor of the photocatalytic activity in this paper the plasmon phenomenon or the polarization electric field effect?
8. Figure 4e shows the SKPFM under dark conditions, and the obtained surface potential of the sample cannot fully prove its efficient charge separation and strong surface electric field. The surface potential difference before and after illumination should be compared.
9. "Angew. Chem., Int. Ed." should be corrected as "Angew. Chem. Int. Ed." Please check whole manuscript in detail to avoid the analogous errors. Also, the format of references should be consistent.

Reviewer #2 (Remarks to the Author):

In this manuscript, the authors report that utilization of ferroelectric polarization electric field as a novel and efficient strategy for forming periodic one-dimensional single-atom arrays on photocatalysts, which allows enhanced charge separation and upgraded reactive sites, finally achieving an enhanced CO₂ photoreduction activity. It is also demonstrated to be general for obtaining different metal single-atom arrays. This finding is interesting and may enlighten new directions for single-atom catalysis and

synthesis of unique nanostructures. Thus, it is worthy of publication under the condition that the following issues are addressed:

1. Investigate whether the polarization electric field can enhance the photocatalytic performance of Au nanoparticles loaded on the polarized sample.
2. Clarify why the performance of BTOAu₄ is lower than that of other samples despite the typical increase in performance with metal loading.
3. Explain the discrepancy in the white lines in XANES for i-SA and SA array, both of which are single atoms.
4. Analyze the catalyst's durability after the photoreaction and include a cycling test of the pure sample to compare its stability with that of the optimal sample.
5. Clarify the key step involved in synthesizing Au i-SA, SA array, and nanoparticles on Bi₄Ti₃O₁₂ photocatalyst.
6. Revise minor errors, such as the annotation in Fig. 3d, which requires revision to improve its quality, and changing the color of the Au atom label in Fig. 5 to improve understanding.

Reviewer #3 (Remarks to the Author):

In "Polarization electric field enabling periodic one-dimensional single-atom arrays on ferroelectric single-crystal nanosheets for CO₂ photoreduction", Lui and coworkers investigate the role of single atom Au arrays on BTO in accelerating the CO₂RR. These arrays are prepared by Corona poling treatment of the BTO nanosheets, which were then used to perform CO₂RR, showing improved result. The ideas presented here are not particularly new, since enhanced photocatalytic activity of nanoparticles subject to Corona poling has been reported recently (see for e.g., Zhang et al, *Ceramics International* 48.11 (2022): 15908-15912). What is new here is to attempt to control the morphology of the Au adsorbates and to characterize them through a comprehensive, if somewhat standard, suite of spectroscopic technique. Further, some QM based calculations are presented to largely confirm the experimental hypothesis. While this work presents some interesting results, I have some conceptual difficulties with the approach and some of the claims that would need to be addressed before it can be published.

1. The authors claim to be able to directly control the 1D structure of SAC using PEF, whoever a similar level of control, albeit using CVD, was recently presented (Guo, *Nat. Synth.* 1, 245-253 (2022)). CVD is a much more accessible and viable approach than PEF, and it's not clear to me what the advantage of the present approach in this context would be.
2. The authors say: "Bi₄Ti₃O₁₂ (BTO) single crystal nanosheets were synthesized by a hydrothermal method". It's unclear to me that this was actually the case, looking at the XRD (which should also be labelled to show the different crystal facets present in the single crystal). This should be confirmed by means of approaches such as reflection high-energy electron diffraction and/or reciprocal space mapping.
3. The authors claim: "The white line of XANES analysis indicates that the valance state of Au SAs on BTOAu₃ and BTOAu are between Au⁰ and Au³⁺, suggesting that Au SAs are positively charge(d) with

partially unoccupied 5s or 4d orbitals (Fig. 2i)". First, this statement is very vague, and is ultimately not very meaningful. Additionally, the statement: "The first peak of BTOAu₃ and BTOPAu matches that of Au₂O₃ reference rather than Au foil, indicating that the stable Au oxidation state excludes the formation of Au clusters or nanoparticles", is entirely unclear from figure 2i, which has pretty poor resolution and should be updated. Also, the corresponding K-edge XAS should be provided to support this point, as is standard in the literature.

4. The statement "Best-fit parameters extracted from the Au L₃-edge FT-EXAFS spectra suggest that BTOPAu shares a high level of similarity in the first shell structure of 1D Au SAs array and perfectly matches the density functional theory (DFT) results (Fig. 2k).", is incorrect. The match is by no means perfect, and the authors are encouraged not to make such statements. The experiment has a lot more structure than the DFT calculation. What is the origin of this, and specifically, why in the 2nd shell peak more intense in the experiment than in the DFT? I would have expected the opposite.

5. The statement "In addition, an obvious surface-enhanced Raman effect has been observed on BTOPAu and BTOAu₃, revealing the interaction between Au SAs and BTO substrate." must be supported by a figure in the supplementary materials, or removed from the text.

6. The DFT calculations, and the succeeding analysis, is severely hampered by the well-known errors in DFT associated with electron delocalization, which is especially acute for strongly correlated systems (Himmetoglu, et al, International Journal of Quantum Chemistry, 114.1 (2014): 14-49), such as the BTO Au considered here. This leads to wildly inaccurate energies. These calculations will need to be redone to be meaningful, at least at the DFT+U level of theory, to conform to the bare minimum of standards in the field.

7. In 5a, according to the authors calculations, they appear to show that the formation energy of a 1D Au array is -76(!) eV with applied voltage, and +1 eV without. These numbers make no sense and need to be placed in proper context. How were these formation energies calculated? How was applied voltage introduced? For the 1D array, are these numbers normalized per atom or not? What is the assumed voltage in these calculations? A 76 eV binding energy is simply not credible

8. Please include the reference BTO Au 4f XPS spectra in figure 5.a in supplementary materials

9. There are several typos that would need to be addressed in the editorial review stage.

Reviewer #4 (Remarks to the Author):

In this manuscript, the authors used the corona poling to regulate the SAs arrangement and create the 1D SAs arrays on ferroelectric BTO oxide nanosheets. The atomic configuration of the 1D SAs arrays provides more adsorption sites and lowers the energy barrier of the rate-determining step. Due to that, the performance of CO₂ reduction to CO is significantly facilitated. However, I believe the manuscript's quality is not suitable for publication in Nature Communications due to insufficient understanding, explanation, and supporting data for the underlying mechanisms for the enhanced PC CO₂ reduction.

1. The corona poling switches and aligns the domains' direction of BTO to provide anchoring sites. But the detailed mechanism is not clear enough for readers. As shown in the schematic of Figure 1a, polarized charge arrays are presented in BTOP along the 1D. Why ?? no explanation for this important

issue.

2. The authors show the TEM image to support the formation of 1D Au array. What is the large area image of the sample before and after poling ?? Since the PC performance was measured under a large area condition, the distribution of 1D Au array, or the coverage of 1D Au array should be provided.

3. For BTO, it should also have intrinsic ferroelectricity. It seems that there is no such observation for 1D Au array on BTO. The difference between BTO and BTOP should be only about the degree or magnitude of polarization.

4. The largest remnant polarization from BTOPAu is shown in Figure 4d. However, why under almost all electric fields, the polarization of BTOPAu is much larger than BTOP? It seems 1D SAs arrays have other influences on polarization. The reason for the suppression of the depolarization effect is attributed to the 1D Au arrays for BTOPAu, but the suppression of the depolarization effect is already quite significant in BTOP if considering by ratio of remnant polarization/maximum polarization.

5. In Figure 4d, why does not the polarization maximum of BTOP and BTOPAu occur under the largest electric fields of 20 kV cm⁻¹? Even the remnant polarization (zero electric field) is larger than those under some electric fields.

6. From the theoretical calculation, the authors revealed the stable formation of the Au array when a voltage is applied. But what the minimum magnitude of the voltage should have? If only talking about the voltage, the BTO with ferroelectricity may also meet this condition. But apparently, the experimental results show BTO cannot form the 1D SAs arrays. Thus, only considering the voltage may not be sufficient to explain the stable formation of 1D SAs arrays.

7. In addition, the authors provide XANES and EXAFS data to support the formation of SACs. However, the explanation of X-ray absorption spectra (including XANES and EXAFS parts) lacks clarity and precision. Since the formation of SACs is the most critical issue in this work, several issues that need to be addressed are as follows:

(i) How do the authors normalize the XANES of Au L3-edge? The normalization appears to be inadequately performed.

(ii) The authors may check carefully if there is a small feature presented at about 11900-11910 eV for BTOAu₃ and BTOPAu. Any interpretation for this small feature?

(iii) How can the authors suggest Au SAs on BTOAu₃ and BTOAu have valence state between Au⁰ and Au³⁺? The author should explain how they obtain the valence state. It seems the authors did not use the common method (energy of absorption edge or peak maximum) to determine the valence state.

(iv) The authors' claim that Au SAs are coordinated with O atoms requires further support. It is recommended that the authors provide the EXAFS fitting results to back up their claim. Without presenting the EXAFS fitting results, it is difficult to justify the coordination environment of Au SAs with O atoms.

(v) Again, why can the authors claim that the second shell observed in BTOPAu is owing to Au-Au coordination without presenting EXAFS fitting results?

(vi) The authors mentioned, "Best-fit parameters extracted from the Au L3-edge FT-EXAFS...". However, no fitting parameters are provided in the manuscript, making it difficult to justify whether the fitting is

reasonable or not. Following up the above point, the statement “Best-fit parameters.....matches the density functional theory” ... How close they are and how they can be compared are not addressed here.

(vii) The fitting data for BTOPAu does not match the experimental data, casting doubt on the presence of a second shell Au-Au.

Reviewer #1:

The manuscript propose that utilizing PEF to induce the synthesis of periodic 1D Au SAs arrays on ferroelectric $\text{Bi}_4\text{Ti}_3\text{O}_{12}$ single-crystal nanosheets through Corona poling followed by ice-bath treatment. The as-prepared poled BTO assembled with 1D SAs arrays exhibits prominent CO_2 photoreduction activity. However, some critical issues should be addressed.

1. The XRD patterns of the reference material (like BTOAuNP) should be given.

Response: Thank you for this comment. The XRD patterns of the reference materials (BTOAuNP and BTOPAuNP) have been supplied in Supplementary Fig. 1 in revised manuscript.

Figure R1 (Supplementary Fig. 1). XRD patterns of BTO, BTOP, BTOAu_X (X=1, 2, 3, 4), BTOPAu, BTOAuNP and BTOPAuNP.

2. The authors claimed that the all samples exhibit a thickness of 5 unit-cell layers. The authors should supplement the related experiments (like AFM) to further study thickness of the samples.

Response: We appreciate this important suggestion. In this work, the samples have a nanosheet morphology and we are fortunate to capture their side view in TEM image. The thickness of BTOPAu is measured to be around 8.2 nm (Figure R2). As the thickness of a single cell layer of BTO from the crystal structure is around 1.6 nm

(Figure R3), it can be concluded that the samples consist of 5 unit-cell layers. For further confirmation, atomic force microscopy (AFM) measurement was conducted. It can be seen that the thickness of BTOPAu varies from ~ 6.2 nm to ~ 10.1 nm (Figure R4), indicating that the sample exhibits an ultrathin structure rather than strictly consisting of 5 unit-cell layers. Consequently, we have revised the statement in the revised manuscript, as follows:

“The scanning electron microscopy (SEM), transmission electron microscopy (TEM) images show that all samples exhibit a sheet-like morphology with clean surface (Fig. 1b-d; Supplementary Fig. 2). The atomic force microscopy (AFM) reveals a range of BTOPAu thickness varying from ~ 6.2 nm to ~ 10.1 nm (Supplementary Fig. 3), indicating that the sample exhibits an ultrathin structure”.

Figure R2 (Fig. 1d). TEM image of BTOPAu.

Figure R3. Crystal structure model of Bi₄Ti₃O₁₂.

Figure R4 (Supplementary Fig. 3). **a** AFM image and **b** corresponding height curve (white narrow in **a**) of BTOPAu.

3. Figure 2d only shows a Au SAs array. Is this random? What is the probability of forming an array? Authors should provide a wider view of the HADDF-STEM images to show that the array formation is not random.

Response: We appreciate this insightful comment. The Au SAs array depicted in Fig. 2d is not random, instead, it originates from an enlarged section of the array A area in Fig. 2c (Figure R5). The Fig. 2c is a wider view image, and the Fig. 2b and Fig. 2d are magnified views from Fig. 2a and Fig. 2c, respectively. Now, we provide the Figure R5 to observe the Au SAs array more clearly. Upon close observation of Figure R5, there are many arrays, more of which are marked now. Though they have different lengths, but their orientations remain approximately consistent. Furthermore, to ensure the reliability and reproducibility of our results, we repeated the HADDF-STEM experiment (Figure R6), further confirming the ubiquitous presence of Au SAs arrays in various areas of BTOPAu.

Figure R5 (Fig. 2c). HADF-STEM image of BTOPAu.

Figure R6. HADF-STEM images of BTOPAu.

4. The signal noise ratio is low in Figure 2c, resulting in the possibility of getting untrusted data. Thus, authors should repeat experiments.

Response: Thanks for this suggestion. Actually, the Au SAs array can be clearly observed on Fig. 2c when we enlarge the image. But we still repeated the HADF-

STEM experiment to avoid errors from untrustworthy data (Figure R6).

5. Figure 2j shows a small peak around 2.90 Å in BTOPAu, corresponding to the second shell Au-Au coordination. But the authors claimed that the the formation of Au-O-Au structure in Au SAs array structure. Why not the Au-O-Bi or Au-O-Ti structure or others? Please add sufficient tests to prove this. Besides, the detailed fitting curves and parameters should be given.

Response: Thanks for your valuable comment. We acknowledge that the fitting in the initial manuscript was somewhat rough, and in response, we have made significant improvements to the fitting curves and parameters in the revised manuscript. The updated data, along with the fitting results, are now presented in Fig. 2j, Supplementary Fig. 6 and Supplementary Table 1. There is a small peak around 2.90 Å in BTOPAu, corresponding to the second shell Au-Au coordination. The fitting parameters from Supplementary Table 1 indicate that the distance of the Au-Au is around 3.524 ± 0.001 Å in *R* space, supporting our claim that the second shell Au-Au coordination arises from the Au SAs array structure. Additionally, the absence of peak around 2.9 Å in BTOAu₃, showing a marked difference from BTOPAu, indicates that this peak does not originate from the Au-O-Bi or Au-O-Ti structures. Considering the evidence presented above, we confidently assert the formation of the Au SAs array structure in BTOPAu. Based on the above new fitting data, we have revised the corresponding descriptions as follows:

“The Fourier-transformed EXAFS (FT-EXAFS) spectra and fitting parameters indicate that Au SAs is coordinated with O atoms on the surface of BTOAu₃ and BTOPAu (Fig. 2j; Supplementary Fig. 6 and Table 1). The distinct Au-O coordination is in stark contrast with the metallic Au-Au coordination observed in Au foil. A small peak around 2.90 Å, corresponding to the second shell Au-Au coordination, is observed in BTOPAu, which is proved by the fitting analysis, confirming the formation of Au-O-Au structure in Au SAs array structure (Fig. 2k).”

Figure R7 (Fig. 2j). Au L_3 -edge EXAFS spectra in R space of BTOAu₃, BTOPAu and references.

Figure R8 (Supplementary Fig. 6). Fitting results of **a-d** the k^3 -weighted FT spectrum in R space and **e-h** $k^3\chi$ data in K space of BTOPAu, BTOAu₃ and references at Au L_3 -edge.

Table R1 (Supplementary Table 1). EXAFS fitting parameters at the Au L_3 -edge for samples.

Sample	Shell	CN^a	$R(\text{\AA})^b$	$\sigma^2(\text{\AA}^2)^c$	$\Delta E_0(\text{eV})^d$	R factor
Au-foil	Au-Au	8*	2.845 ± 0.019	0.0151	6.7	0.0032
	Au-Au	6*	2.855 ± 0.009	0.0050	3.1	
Au ₂ O ₃	Au-O	4.1 ± 0.5	1.986 ± 0.001	0.0054	10.6	0.0124
BTOAu ₃	Au-O	3.8 ± 0.2	1.964 ± 0.007	0.0074	7.4	0.0098
BTOPAu	Au-O	1.9 ± 0.1	2.035 ± 0.001	0.0016	13.1	0.0180
	Au-Au	4.4 ± 0.9	3.524 ± 0.001	0.0010	3.2	

^a CN , coordination number; ^b R , the distance to the neighboring atom; ^c σ^2 , the Mean Square Relative Displacement (MSRD); ^d ΔE_0 , inner potential correction; R factor indicates the

goodness of the fit. S_0^2 was fixed to 0.856, according to the experimental EXAFS fit of Au foil by fixing CN as the known crystallographic value. * This value was fixed during EXAFS fitting, based on the known structure of Au. Fitting range: $3.0 \leq k (\text{\AA}^{-1}) \leq 12.0$ and $1.0 \leq R (\text{\AA}) \leq 3.3$ (Au foil); $3.0 \leq k (\text{\AA}^{-1}) \leq 12.0$ and $1.0 \leq R (\text{\AA}) \leq 2.0$ (Au_2O_3); $1.5 \leq k (\text{\AA}^{-1}) \leq 8.0$ and $1.0 \leq R (\text{\AA}) \leq 2.2$ (BTOAu_3); $2.5 \leq k (\text{\AA}^{-1}) \leq 7.8$ and $1.1 \leq R (\text{\AA}) \leq 3.2$ (BTOPAu). A reasonable range of EXAFS fitting parameters: $0.700 < S_0^2 < 1.000$; $CN > 0$; $\sigma^2 > 0 \text{\AA}^2$; $|E_0| < 15 \text{ eV}$; $R \text{ factor} < 0.02$.

6. The authors claimed that the Pt, Fe, Co, Ni and Ag metal atoms are dispersedly anchored on the BTO surface to form i-SAs. However, the Supplementary Fig. 7 can not distinguish between different metal i-SAs. Please add the related experiments (like EDS) to verify the different metal elements on the BTO surface.

Response: Thanks so much for this great suggestion, which is very helpful to identify the different metals accurately. Initially, we attempted to conduct element mapping at the atomic level (Figure R9), but encountered challenges due to the micro vibrations of samples under the electron beam of TEM. As a result, obtaining useful data from this approach is difficult.

To address this limitation, we made modifications in our experimental approach. When collecting the HADDF-STEM images of metals (Pt, Fe, Co, Ni, and Ag) loaded on both BTO and BTOP (Supplementary Fig. 9), we simultaneously gathered corresponding element mapping images over a larger area. This approach allowed us to obtain more comprehensive information and better verify the presence of different metal elements. These supplementary element mapping images are now included in Supplementary Fig. 10. Thanks for this comment again!

Figure R9. Atomic element mapping of BTOAu.

Figure R10 (Supplementary Fig. 10). Element mapping corresponding to the metal SAs (Pt, Fe, Co, Ni and Ag) loaded on **a-e** BTO and **f-j** BTOP.

7. As we all know, in the presence of Au, light will induce the surface plasmon field, so is the dominant factor of the photocatalytic activity in this paper the plasmon phenomenon or the polarization electric field effect?

Response: Thanks for this comment. We agree that light-induced surface plasmon fields can indeed influence the photocatalytic activity in the presence of Au. In this

work, we synthesized various samples, including BTOAu₃, BTOPAu, BTOAuNP, and BTOPAuNP, all of which exhibit surface plasmon phenomenon. However, the photocatalytic performance of BTOPAu and BTOAu₃ suppresses that of the other samples, suggesting that the plasmon phenomenon is not the dominant factor.

Besides, for BTOAu₃ and BTOPAu samples with the similar Au loading amount based on ICP results (Supplementary Table 4), both experiments (photoelectrochemical analysis) and DFT calculation indicated that BTOPAu exhibits more favorable charge separation (Fig. 4a, b) and lower CO₂ activation barriers (DFT calculation, Fig. 5g) compared to BTOAu₃. These observations were attributed to the polarization electric field effect, which forms Au SAs array on BTO, suggesting that this is the dominant factor contributing to the enhanced photocatalytic activity in our study.

8. Figure 4e shows the SKPFM under dark conditions, and the obtained surface potential of the sample cannot fully prove its efficient charge separation and strong surface electric field. The surface potential difference before and after illumination should be compared.

Response: Thank you for this constructive suggestion. To demonstrate the charge transfer ability under photoexcitation, we have supplemented SKPFM before and after illumination in revised manuscript (Fig. 4e-f), as follows:

“Before illumination, both BTOP, BTOAu₃ and BTOPAu show larger surface potential compared to BTO, indicating that the presence of polarization electric field and Au SAs enhance the surface electric field of BTO. After illumination, all samples exhibit an elevation in surface potential compared to that in their dark state. This change indicates an increase in charge separation and redistribution under photoexcitation for all the synthesized samples. The difference of surface potential ($\Delta\Phi$) between in before and after illumination is defined as follows:

$$\Delta\Phi = \Delta_{ave(illumination)} - \Delta_{ave_{dark}}$$

Here $\Delta\Phi$ is determined to be 10 mV for BTO, 25 mV for BTOP, 13 mV for

BTOAu₃ and 35 mV for BTOPAu. It confirms that both polarization electric field and Au SAs loading play important roles in promoting charge separation. The largest $\Delta\Phi$ observed in BTOPAu indicate the formation of the most robust electric field, which is attributed to the synergistic effect of the polarization electric field and Au SAs loading, thus largely facilitating the rapid separation and transfer of photogenerated charges and contributing to the enhanced photocatalytic activity of BTOPAu.”

Figure R11 (Fig. 4e-f). Surface potential distribution (corresponding to BTO, BTOP, BTOAu₃ and BTOPAu **a**₁-**a**₄ before and **b**₁-**b**₄ after illumination, respectively; inset: AFM images; Δ_{ave} defined as absolute value of surface average potential).

9. “Angew. Chem., Int. Ed.” should be corrected as “Angew. Chem. Int. Ed.” Please check whole manuscript in detail to avoid the analogous errors. Also, the format of references should be consistent.

Response: We are sorry for the slip. This manuscript has been carefully checked and revised to avoid such errors. Thanks very much again for all your comments and suggestions above.

Reviewer #2:

In this manuscript, the authors report that utilization of ferroelectric polarization electric field as a novel and efficient strategy for forming periodic one-dimensional single-atom arrays on photocatalysts, which allows enhanced charge separation and upgraded reactive sites, finally achieving an enhanced CO₂ photoreduction activity. It is also demonstrated to be general for obtaining different metal single-atom arrays. This finding is interesting and may enlighten new directions for single-atom catalysis and synthesis of unique nanostructures. Thus, it is worthy of publication under the condition that the following issues are addressed:

1. Investigate whether the polarization electric field can enhance the photocatalytic performance of Au nanoparticles loaded on the polarized sample.

Response: Thanks so much for this suggestion. We have loaded Au nanoparticles on BTOP to analyze whether the polarization electric field can enhance the performance of Au nanoparticles. The synthetic process is as following: 0.10 g of BTOP sample powder and 0.2 mL 1 g/100 mL HAuCl₄ solution are mixed in 30 mL deionized water. To achieve an adsorption-desorption equilibrium, the suspension is stirred in darkness for 30 min and then irradiated by a 300 W mercury lamp with continuous stirring. Then the sample was washed with deionized water and dried at -50 °C for 12 h to obtain BTOP loaded with Au nanoparticles (BTOPAuNP). The CO evolution of BTOPAuNP was determined to be 34.10 μmol·g⁻¹ within 4 h, a value similar to that of BTOAuNP (34.86 μmol·g⁻¹) (Supplementary Fig. 11). Therefore, it can be concluded that the polarization electric field did not enhance the photocatalytic CO₂ reduction performance of BTOPAuNP. The mechanism responsible for the improved performance of Au SAs on BTOP lies in that the polarization electric field alters the permutations of i-SAs to periodic Au SAs array consisting of neighboring Au catalytic sites. However, in the case of Au nanoparticles, the catalytic sites become aggregated, leading to a significant reduction in the availability of active sites and consequently diminishing the overall catalytic efficiency. Unfortunately, the polarization electric field is unable to change the accumulation form of massive atoms in nanoparticles. Consequently, its capacity to enhance the catalytic performance of Au nanoparticles is

limited.

Figure R12 (Supplementary Fig. 11). CO production from CO₂ photoreduction over BTOAuNP and BTOPAuNP for 4 h under simulated solar light.

2. Clarify why the performance of BTOAu₄ is lower than that of other samples despite the typical increase in performance with metal loading.

Response: Thank you for this comment. The catalytic performance exhibits a gradual improvement with increasing Au loading, observed from BTOAu₁ to BTOAu₂ to BTOAu₃, with exception of BTOAu₄. This trend may be due to a small amount of Au atom agglomeration (cluster or nanoparticle) in BTOAu₄, leading to insufficient exposure of the catalytic sites and consequently a decrease in catalytic activity compared to BTOAu₃. This observation also underscores the importance of maintaining an optimal balance in Au loading amount to maximize the catalytic efficiency.

3. Explain the discrepancy in the white lines in XANES for i-SA and SA array, both of which are single atoms.

Response: Thanks for this comment. The discrepancy in white lines intensity is attributed to this difference to the variation in the Au valence state between BTOAu₃ and BTOPAu. In BTOAu₃, the coordination number of Au i-SAs is 3.8 ± 0.2 , while in

BTOPAu, it is found to be 1.9 ± 0.1 (Table R2). As a result, the Au oxidation state in BTOAu₃ is higher compared to BTOPAu, leading to a higher white line intensity in BTOAu₃ compared to BTOPAu.

Table R2 (Supplementary Table 1). EXAFS fitting parameters at the Au L_3 -edge for samples.

Sample	Shell	CN^a	$R(\text{\AA})^b$	$\sigma^2(\text{\AA}^2)^c$	$\Delta E_0(\text{eV})^d$	R factor
Au-foil	Au-Au	8*	2.845 ± 0.019	0.0151	6.7	0.0032
	Au-Au	6*	2.855 ± 0.009	0.0050	3.1	
Au ₂ O ₃	Au-O	4.1 ± 0.5	1.986 ± 0.001	0.0054	10.6	0.0124
BTOAu ₃	Au-O	3.8 ± 0.2	1.964 ± 0.007	0.0074	7.4	0.0098
BTOPAu	Au-O	1.9 ± 0.1	2.035 ± 0.001	0.0016	13.1	0.0180
	Au-Au	4.4 ± 0.9	3.524 ± 0.001	0.0010	3.2	

^a CN , coordination number; ^b R , the distance to the neighboring atom; ^c σ^2 , the Mean Square Relative Displacement (MSRD); ^d ΔE_0 , inner potential correction; R factor indicates the goodness of the fit. S_0^2 was fixed to 0.856, according to the experimental EXAFS fit of Au foil by fixing CN as the known crystallographic value. * This value was fixed during EXAFS fitting, based on the known structure of Au. Fitting range: $3.0 \leq k (\text{\AA}^{-1}) \leq 12.0$ and $1.0 \leq R (\text{\AA}) \leq 3.3$ (Au foil); $3.0 \leq k (\text{\AA}^{-1}) \leq 12.0$ and $1.0 \leq R (\text{\AA}) \leq 2.0$ (Au₂O₃); $1.5 \leq k (\text{\AA}^{-1}) \leq 8.0$ and $1.0 \leq R (\text{\AA}) \leq 2.2$ (BTOAu₃); $2.5 \leq k (\text{\AA}^{-1}) \leq 7.8$ and $1.1 \leq R (\text{\AA}) \leq 3.2$ (BTOPAu). A reasonable range of EXAFS fitting parameters: $0.700 < S_0^2 < 1.000$; $CN > 0$; $\sigma^2 > 0 \text{\AA}^2$; $|\Delta E_0| < 15 \text{ eV}$; R factor < 0.02 .

4. Analyze the catalyst's durability after the photoreaction and include a cycling test of the pure sample to compare its stability with that of the optimal sample.

Response: Thanks for your suggestion. The XRD patterns of BTO and BTOPAu₃ have been analyzed before and after photocatalytic reaction (Supplementary Fig. 12b). There is no any observable change in XRD patterns, demonstrating the structural stability during photocatalytic reaction. The cycling test of BTO sample has been supplied (Supplementary Fig. 12a). Both BTO and BTOPAu₃ exhibit consistently photocatalytic activity and high stability throughout the cycling test, affirming their excellent durability.

Figure R13 (Supplementary Fig. 12). **a** Cycling test of photoreduction CO₂ reduction into CO over BTO and BTOPAu. **b** XRD patterns of BTO and BTOPAu before and after reaction.

5. Clarify the key step involved in synthesizing Au i-SA, SA array, and nanoparticles on Bi₄Ti₃O₁₂ photocatalyst.

Response: Thanks for this comment. The synthesis of Au nanoparticles in this work involves the photodeposition method, wherein the critical step is that the CB of semiconductor is more negative than the reduction potential of Au³⁺ to Au⁰. For BTO and BTOP, their CB values are approximately -0.93 and -0.83 (V vs. NHE, pH=0), respectively, both of which are more negative than the reduction potential of Au³⁺ to Au⁰ (+1.002 V vs. NHE, pH=0).^{1,2} Thus, Au nanoparticles can effectively form on the reduction reaction sites of BTO and BTOP by the photogenerated electrons. For the synthesis of Au i-SAs in this work, the key step involves creating an ice bath environment. The metal precursor is frozen to ice using liquid nitrogen, and then Au ions are gradually released during the ice melting process, in which the Au ions adsorbed on the surface of BTO are reduced to be Au i-SAs.³ Regarding the formation of 1D Au SAs array, the crucial step is the introduction of ferroelectric polarization electric field, which can allow the aligned arrangement of domains of BTO, resulting in the periodic permutations of polarized positive and negative charges. During this process, the Au precursor orderly adsorbs on the surface of BTOP, and subsequently is reduced to form the 1D Au SAs array.

6. Revise minor errors, such as the annotation in Fig. 3d, which requires revision to improve its quality, and changing the color of the Au atom label in Fig. 5 to improve understanding.

Response: We are sorry for these slips. The annotation in Fig. 3d and the color of the Au atom label in Fig. 5 have been revised. Thanks very much again for all your comments and suggestions above.

Reviewer #3:

In "Polarization electric field enabling periodic one-dimensional single-atom arrays on ferroelectric single-crystal nanosheets for CO₂ photoreduction", Lui and coworkers investigate the role of single atom Au arrays on BTO in accelerating the CO₂RR. These arrays are prepared by Corona poling treatment of the BTO nanosheets, which were then used to perform CO₂RR, showing improved result. The ideas presented here are not particularly new, since enhanced photocatalytic activity of nanoparticles subject to Corona poling has been reported recently (see for e.g., Zhang et al, *Ceramics International* 48.11 (2022): 15908-15912). What is new here is to attempt to control the morphology of the Au adsorbates and to characterize them through a comprehensive, if somewhat standard, suite of spectroscopic technique. Further, some QM based calculations are presented to largely confirm the experimental hypothesis. While this work presents some interesting results, I have some conceptual difficulties with the approach and some of the claims that would need to be addressed before it can be published.

1. The authors claim to be able to directly control the 1D structure of SAC using PEF, whoever a similar level of control, albeit using CVD, was recently presented (Guo, *Nat. Synth.* 1, 245-253 (2022)). CVD is a much more accessible and viable approach than PEF, and it's not clear to me what the advantage of the present approach in this context would be.

Response: Thank you for this valuable suggestion. CVD is indeed a practical and widely-used technique for thin-film, particularly on the flat and smooth surfaces. However, it may present challenges in achieving uniform atomic layers on uneven surfaces or cracks. In our work, the PEF technology is employed to achieve the precise control synthesis of single atom arrays on powder samples. This is very meaningful since most of catalysts are commonly made in powdery form. Furthermore, PEF has the potential to be employed on ferroelectric thin films and the materials capable of producing dipoles. This versatility allows its application to a wide range of materials in various fields, including catalysis, electronics, piezoelectric devices, capacitors, microelectromechanical systems, and electro-optics.

2. The authors say: "Bi₄Ti₃O₁₂ (BTO) single crystal nanosheets were synthesized by a hydrothermal method". It's unclear to me that this was actually the case, looking at the XRD (which should also be labelled to show the different crystal facets present in the single crystal). This should be confirmed by means of approaches such as reflection high-energy electron diffraction and/or reciprocal space mapping.

Response: Thanks so much for this important suggestion. We have incorporated the labeling of different crystal facets in the revised manuscript (Figure R14). Strictly speaking, the BTO nanosheet is microscopic-size single crystal, while reflection high-energy electron diffraction and reciprocal space mapping are more suitable for single crystal films or macroscopic single crystals. Thus, SAED analysis was conducted. As shown in the inset of Fig. 1b, the SAED pattern of a single BTO nanosheet reveals only one set of diffracted spots instead of multiple sets of diffracted spots with rotational relationships.⁴ As the sample is in powder form, in XRD measurement, numerous nanosheets are randomly stacked together, showcasing different crystal facets. Based on the results from SAED pattern, it can be asserted that the BTO nanosheet is indeed single crystal.

Figure R14 (Supplementary Fig. 1). XRD patterns of BTO, BTOP, BTOAu_x (X=1, 2, 3, 4), BTOPAu, BTOAuNP and BTOPAuNP.

Figure R15 (Fig. 1b). TEM image and SAED pattern (inset).

3. The authors claim: "The white line of XANES analysis indicates that the valence state of Au SAs on BTOAu₃ and BTOPAu are between Au⁰ and Au³⁺, suggesting that Au SAs are positively charge(d) with partially unoccupied 5s or 4d orbitals (Fig. 2i)". First, this statement is very vague, and is ultimately not very meaningful. Additionally, the statement: "The first peak of BTOAu₃ and BTOPAu matches that of Au₂O₃ reference rather than Au foil, indicating that the stable Au oxidation state excludes the formation of Au clusters or nanoparticles", is entirely unclear from figure 2i, which has pretty poor resolution and should be updated. Also, the corresponding K-edge XAS should be provided to support this point, as is standard in the literature.

Response: We are thankful for this constructive comment. We have made the updates to the XANES curves and EXAFS data (fitting curves and parameters) in the revised manuscript. Additionally, we have adjusted the image resolution and line width to ensure clear and accurate images. For the *L*-edge XAS data, we initially used the white line intensity to reflect the corresponding valence states of elements (Figure R16). However, we agree that this statement may not be meaningful. Consequently, we have decided to remove this depiction from the revised manuscript. For the Au *K*-edge adsorption, its energy is around 80725 eV, which is very challenging for analysis, leading most prior research works to focusing on the Au *L*-edge data.^{5,6} Although we attempted to obtain Au *K*-edge data from several synchrotron light source centers, unfortunately, we did not receive a positive response. As an alternative, we conducted

EXAFS data fitting to analyze the Au SAs coordination and configuration on BTOAu₃ and BTOPAu (Figure R17 and Table R1). The results show that the Au first shell is Au-O structure in both BTOAu₃ and BTOPAu, and the second shell of Au SAs in BTOPAu is Au-Au, confirming the presence of the Au i-SAs in BTOAu₃ and Au SAs array structure in BTOPAu. Now, we have revised the corresponding descriptions in the revised manuscript, as follows:

“The white line intensity of XANES analysis indicates that the stable Au oxidation state excludes the formation of Au clusters or nanoparticles, where the peak intensity of BTOAu₃ and BTOPAu are between Au foil and Au₂O₃ reference (**Fig. 2i**).^{20,21} The EXAFS data and fitting analysis indicate the coordination environment of Au SAs on BTOAu₃ and BTOPAu. The Fourier-transformed EXAFS (FT-EXAFS) spectra and fitting parameters indicate that Au SAs is coordinated with O atoms on the surface of BTOAu₃ and BTOPAu (**Fig. 2j**; **Supplementary Fig. 6 and Table 1**). The distinct Au-O coordination is in sharp contrast with the metallic Au-Au coordination observed in Au foil. A small peak around 2.90 Å, corresponding to the second shell Au-Au coordination, is observed in BTOPAu, which is proved by the fitting analysis, confirming the formation of Au-O-Au structure in Au SAs array structure (**Fig. 2k**).”

We appreciate your valuable comment, which guides us to enhance the presentation of our findings.

Figure R16 (Fig. 2j). Au L_3 -edge normalized XANES spectra of BTOAu₃, BTOPAu and references.

Figure R17 (Supplementary Fig. 6). Fitting results of **a-d** the k^3 -weighted FT spectrum in R space and **e-h** $k^3\chi$ data in K space of BTOPAu, BTOAu₃ and references at Au L_3 -edge.

Table R3 (Supplementary Table 1). EXAFS fitting parameters at the Au L_3 -edge for samples.

Sample	Shell	CN^a	$R(\text{\AA})^b$	$\sigma^2(\text{\AA}^2)^c$	$\Delta E_0(\text{eV})^d$	R factor
Au-foil	Au-Au	8*	2.845 ± 0.019	0.0151	6.7	0.0032
	Au-Au	6*	2.855 ± 0.009	0.0050	3.1	
Au ₂ O ₃	Au-O	4.1 ± 0.5	1.986 ± 0.001	0.0054	10.6	0.0124
BTOAu ₃	Au-O	3.8 ± 0.2	1.964 ± 0.007	0.0074	7.4	0.0098
BTOPAu	Au-O	1.9 ± 0.1	2.035 ± 0.001	0.0016	13.1	0.0180
	Au-Au	4.4 ± 0.9	3.524 ± 0.001	0.0010	3.2	

^a CN , coordination number; ^b R , the distance to the neighboring atom; ^c σ^2 , the Mean Square Relative Displacement (MSRD); ^d ΔE_0 , inner potential correction; R factor indicates the goodness of the fit. S_0^2 was fixed to 0.856, according to the experimental EXAFS fit of Au foil by fixing CN as the known crystallographic value. * This value was fixed during EXAFS fitting, based on the known structure of Au. Fitting range: $3.0 \leq k (\text{\AA}^{-1}) \leq 12.0$ and $1.0 \leq R (\text{\AA}) \leq 3.3$ (Au foil); $3.0 \leq k (\text{\AA}^{-1}) \leq 12.0$ and $1.0 \leq R (\text{\AA}) \leq 2.0$ (Au₂O₃); $1.5 \leq k (\text{\AA}^{-1}) \leq 8.0$ and $1.0 \leq R (\text{\AA}) \leq 2.2$ (BTOAu₃); $2.5 \leq k (\text{\AA}^{-1}) \leq 7.8$ and $1.1 \leq R (\text{\AA}) \leq 3.2$ (BTOPAu). A reasonable range of EXAFS fitting parameters: $0.700 < S_0^2 < 1.000$; $CN > 0$; $\sigma^2 > 0 \text{\AA}^2$; $|\Delta E_0| < 15 \text{ eV}$; R factor < 0.02 .

4. The statement "Best-fit parameters extracted from the Au L_3 -edge FT-EXAFS spectra suggest that BTOPAu shares a high level of similarity in the first shell

structure of 1D Au SAs array and perfectly matches the density functional theory (DFT) results (Fig. 2k).", is incorrect. The match is by no means perfect, and the authors are encouraged not to make such statements. The experiment has a lot more structure than the DFT calculation. What is the origin of this, and specifically, why in the 2nd shell peak more intense in the experiment than in the DFT? I would have expected the opposite.

Response: Great thanks for this constructive comment. We acknowledge that the fitting in the initial manuscript was not perfect and somewhat rough. As a result, we have made significant improvements to the fitting curves and parameters in the revised manuscript. The updated data are now presented in Figure R17 and Table R3 (as shown in the response above). We have conducted a detailed analysis of the Au SAs coordination and configuration on BTOAu₃ and BTOPAu. The fitting parameters show the first-shell of Au is Au-O structure on both BTOAu₃ and BTOPAu, and the second-shell of Au is Au-Au on BTOPAu, where the $R(\text{\AA})$ reflects the bond length of Au-O in R space, the $\sigma^2(\text{\AA}^2)^c$, $\Delta E_0(\text{eV})$ and R factor reflect the fitting error, both of them show a reasonable fitting. Thus, we created a rational model for DFT calculations based on the XAS results. These efforts have enabled us to obtain a more comprehensive understanding of the different Au SAs structures in our work.

We apologize for any confusion caused by the previous unclear depiction in the original manuscript. In the revised manuscript, we have made the necessary adjustments to provide a clearer and more accurate statement. The revised depiction now reads as follows:

“The EXAFS data and fitting analysis indicate the coordination environment of Au SAs on BTOAu₃ and BTOPAu. The Fourier-transformed EXAFS (FT-EXAFS) spectra and fitting parameters indicate that Au SAs is coordinated with O atoms on the surface of BTOAu₃ and BTOPAu (**Fig. 2j; Supplementary Fig. 6 and Table 1**). The distinct Au-O coordination is in stark contrast with the metallic Au-Au coordination observed in Au foil. A small peak around 2.90 Å, corresponding to the second shell Au-Au coordination, is observed in BTOPAu, which is proved by the fitting analysis, confirming the formation of Au-O-Au structure in Au SAs array structure (**Fig. 2k**).”

We sincerely appreciate your constructive and important suggestion, which has significantly enhanced the quality of our work.

5. The statement "In addition, an obvious surface-enhanced Raman effect has been observed on BTOPAu and BTOAu₃, revealing the interaction between Au SAs and BTO substrate." must be supported by a figure in the supplementary materials, or removed from the text.

Response: Thanks for this important suggestion. Initially, we referred to prior research on surface-enhanced Raman scattering spectroscopy via Au, which demonstrated that Au exhibits a surface-enhanced Raman effect. In our work, two peaks around 100 and 530 cm⁻¹ were observed, with the peak intensity in BTOPAu stronger than in BTO. We attributed this difference to the potential surface-enhanced Raman effect caused by Au. However, after carefully considering your suggestion, we agree that it may indeed raise some controversies to say "an obvious surface-enhanced Raman effect" based on the observed enhancement of only two peaks and its occurrence in only one sample. Now, we have removed this statement from the revised manuscript. We appreciate your comment!

6. The DFT calculations, and the succeeding analysis, is severely hampered by the well-known errors in DFT associated with electron delocalization, which is especially acute for strongly correlated systems (Himmetoglu, et al, International Journal of Quantum Chemistry, 114.1 (2014): 14-49), such as the BTO Au considered here. This leads to wildly inaccurate energies. These calculations will need to be redone to be meaningful, at least at the DFT+U level of theory, to conform to the bare minimum of standards in the field.

Response: We are grateful for your insightful comment, which has significantly aided in rectifying the deviations observed in our theoretical calculations, particularly pertaining to the formation energy. We have re-conducted all the calculations at the DFT+U level of theory to ensure the accuracy of theoretical simulation results in our revised manuscript. Your contribution is greatly appreciated once more.

7. In 5a, according to the authors calculations, they appear to show that the formation energy of a 1D Au array is -76(!) eV with applied voltage, and +1 eV without. These numbers make no sense and need to be placed in proper context. How were these formation energies calculated? How was applied voltage introduced? For the 1D array, are these numbers normalized per atom or not? What is the assumed voltage in these calculations? A 76 eV binding energy is simply not credible.

Response: Thanks for this constructive comment. We apologize for this rough and inaccurate result. We have conducted a thorough reevaluation of the calculations by employing DFT+U with $U_{eff} = 4.2$ eV.⁷ And we introduced the applied voltage through the INCAR parameters to simulate Corona poling treatment (LDIPOL=.T.; IDIPOL=3; EFIELD=9E-24) and calculated formation energy by the following formula (in Method section):

The formation energy (E_f) was calculated by the followed formula:

$$E_f = E_{tot} - E_{sub} - n\mu_{Au}$$

Where the E_{tot} and E_{sub} are the total energy of the substrate structure with Au loading and the substrate structure, respectively, the n is the number of Au atom, and the μ_{Au} is the energy of one Au atom. As a result, the formation energy of $\text{Bi}_4\text{Ti}_3\text{O}_{12}$ with 1D Au array is calculated to be ~ -18.25 eV with applied voltage and -0.20 eV without applied voltage, as illustrated in Fig. 5a.

The corresponding depiction has now revised, as follows: “Theoretical simulation demonstrates that the Au i-SAs can spontaneously form stable structure on the surface of $\text{Bi}_4\text{Ti}_3\text{O}_{12}$, while the loading of three neighboring Au SAs (imitated 1D Au SAs array) on $\text{Bi}_4\text{Ti}_3\text{O}_{12}$ is thermodynamically unfavorable compared to Au i-SAs. In contrast, the formation of 1D Au SAs array turns out highly spontaneous and stable when a voltage is applied (**Fig. 5a**).”

Figure R18 (Fig. 5a). Theoretically calculated formation energy of isolated Au atom on $\text{Bi}_4\text{Ti}_3\text{O}_{12}$, Au array on $\text{Bi}_4\text{Ti}_3\text{O}_{12}$ without and with applied voltage.

8. Please include the reference BTO Au 4f XPS spectra in figure 5.a in supplementary materials.

Response: Thank you for this kind comment. The BTO Au 4f XPS spectrum has been supplied in Supplementary Fig. 7a, and there is no any Au 4f peak in this BTO spectra.

Figure R19 (Supplementary Fig. 7a). Au 4f of BTO, BTOP, BTOAu₃ and BTOPAu.

9. There are several typos that would need to be addressed in the editorial review

stage.

Response: We are sorry for these oversights. The manuscript has been carefully checked and revised to correct the grammar and typographical errors. Thanks very much again for all your comments and suggestions above.

Reviewer #4:

In this manuscript, the authors used the Corona poling to regulate the SAs arrangement and create the 1D SAs arrays on ferroelectric BTO oxide nanosheets. The atomic configuration of the 1D SAs arrays provides more adsorption sites and lowers the energy barrier of the rate-determining step. Due to that, the performance of CO₂ reduction to CO is significantly facilitated. However, I believe the manuscript's quality is not suitable for publication in Nature Communications due to insufficient understanding, explanation, and supporting data for the underlying mechanisms for the enhanced PC CO₂ reduction.

1. The Corona poling switches and aligns the domains' direction of BTO to provide anchoring sites. But the detailed mechanism is not clear enough for readers. As shown in the schematic of Figure 1a, polarized charge arrays are presented in BTOP along the 1D. Why ?? no explanation for this important issue.

Response: Thank you for this important comment. In this study, the orientation of dipoles plays a crucial role in forming SAs array, and the dipoles align along the domains' direction. Our approach involved utilizing these oriented dipoles to anchor Au SAs through electrostatic adsorption.

For BTO, which is an intrinsic ferroelectric material, it also exhibits domains with Moire fringes. However, the surface properties of ferroelectrics are affected by surface screening fields. These fields absorb ions or charged particles from the environment, leading to a weakening of the polarity of the material. Consequently, the Au anchoring sites are disrupted, resulting in random anchoring on the BTO surface. When Corona poling was applied, the domains' direction in BTO was switched to

break the surface screening effect. This process endows the domains in BTOP with charged properties, which offer anchoring sites along domains, ultimately leading to the formation of well-organized 1D array structure in BTOPAu. The success of this experiment can be attributed to the Corona poling treatment applied before Au loading, which effectively facilitated the desired anchoring behavior. The Corona poling treatment played a crucial role in achieving the desired 1D SAs array structure and provides some bases for future related researches in this area. Now, we have supplemented the corresponding explanations in the Introduction section, as follows:

“Corona poling switches domains’ direction in BTO while breaking the surface screening field, which endows the domains in BTOP with charged properties to anchor Au SAs along domains through electrostatic adsorption, playing a crucial role in synthesis of 1D SAs arrays.”

2. The authors show the TEM image to support the formation of 1D Au array. What is the large area image of the sample before and after poling?? Since the PC performance was measured under a large area condition, the distribution of 1D Au array, or the coverage of 1D Au array should be provided.

Response: Thanks for this great suggestion. It is important to note that though HADDF-STEM provides high-resolution images with valuable information within a fixed amplification range, obtaining useful information for SAs under very large areas is challenging. However, despite the difficulties, we made efforts to present both large area (Fig. 2a: i-SAs; Fig. 2c: 1D SAs arrays) and small area images (Fig. 2b: i-SAs; Fig. 2d: 1D SAs arrays) before and after Corona poling treatment. Specifically, Fig. 2a shows the i-SAs in a large area, and Fig. 2c illustrates the 1D SAs arrays also in a large area. And Fig. 2b and Fig. 2d are enlarged versions of Fig. 2a and Fig. 2c, respectively, providing a more detailed view. The resolution of Fig. 2c is low. Upon zooming in the 1D SAs array, Fig. 2d reveals a clear representation of the array. As we continued to zoom out in Fig. 2a and Fig. 2c, we found that obtaining useful information about atoms became increasingly challenging. Consequently, it is difficult to provide larger area images than what is presented in Fig. 2c. Herein, the

distribution of 1D Au arrays has been framed up in Figure R20 (Fig. 2c). However, this manuscript focuses on a novel method for achieving a 1D SAs array structure and providing an in-depth understanding of the mechanisms behind its improved performance. As the coverage is difficult to be calculated and is not entirely accurate, and as a result, we have chosen not to include this specific coverage in the revised manuscript. Thanks for your understanding!

Figure R20 (Fig. 2c in revised manuscript). HADDF-STEM image of BTOPAu.

3. For BTO, it should also have intrinsic ferroelectricity. It seems that there is no such observation for 1D Au array on BTO. The difference between BTO and BTOP should be only about the degree or magnitude of polarization.

Response: Thank you for this comment. Indeed, BTO is an intrinsic ferroelectric, which also exhibits ferroelectricity and domains. However, the surface properties of ferroelectrics are affected by surface screening fields, where the surface can absorb ions or charged particles from the surrounding environment. As a result, the surface polarity of BTO is weakened, leading to the disruption of Au anchoring sites along the domains. It thus causes random anchoring sites on the surface of BTO. When the

Corona poling was implemented, the domains' direction in BTO was switched to mitigate the surface screening effect. This Corona poling process imparts charged properties to the domains in BTOP, thereby providing anchoring sites along the domains. From a ferroelectricity standpoint, there may not be any obvious difference between BTO and BTOP. However, in terms of dipole distribution, which plays a crucial role in providing sites for Au absorption, there is a significant contrast between the two materials. The results obtained from SKPFM further support this observation. It indicates that the surface electric field of BTOP has been significantly enhanced after the Corona poling treatment compared to BTO (Fig. 4e-f in revised manuscript).

4. The largest remnant polarization from BTOPAu is shown in Figure 4d. However, why under almost all electric fields, the polarization of BTOPAu is much larger than BTOP? It seems 1D SAs arrays have other influences on polarization. The reason for the suppression of the depolarization effect is attributed to the 1D Au arrays for BTOPAu, but the suppression of the depolarization effect is already quite significant in BTOP if considering by ratio of remnant polarization/maximum polarization.

Response: Thank you for this valuable and constructive comment. The electric hysteresis loop in the original manuscript may suffer from inaccuracies due to the sample preparation (sample density) and equipment issues. For ensuring accuracy of data, we have repeated this experiment and updated electric hysteresis loop analysis in the revised manuscript (Fig. 4d).

It can be observed that the Au SAs effectively suppress the depolarization effect in both BTOAu₃ and BTOPAu samples. Notably, the P_r of BTOPAu is slightly higher than that of BTOAu₃. However, the Corona poling treatment almost did not exhibit any significant effect on increasing the P_r for BTOP samples compared to BTO. Nevertheless, we have taken care to verify these results by repeating the electric hysteresis loop experiments (Figure R22), and the trend remains consistent, confirming the observed outcomes. The increased maximum polarization in both BTOP and BTOPAu samples compared to BTO and BTOAu₃ is a direct result of the Corona poling pre-switching the domains within the samples, leading to aligned

arrangement of domains. Additionally, the presence of the Au SAs contributes to an increase in the P_r , with different Au structures exhibiting varying degrees of this effect. Regarding the remnant polarization/maximum polarization ratio, we have not considered it in our work. This is because the maximum polarization dissipates immediately upon removing the external voltage, and over time, the dipoles on the sample's surface neutralize, rendering them unable to adsorb metal ions. The introduction of Au has a dual functions, namely providing surface catalytic sites and maximizing the surface electric field for efficient carrier separation. Now, we have revised the corresponding depiction in revised manuscript, as follows:

“To get a deep insight into the bulk charge separation, electric hysteresis loops are surveyed. BTOAu₃ and BTOPAu shows an obviously larger remnant polarization (P_r) than BTO, which is attributed to that Au SAs loading anchors the domains and keeps them in a non-offset polarization state to a certain extent (**Fig. 4d**).²⁷ The fascinating phenomenon may be attributed that Au SAs arrays anchored by PEF plays a role of pinning effect to inhibit the domains reversal back into disorder state after withdrawing the applied voltage, which can keep a strong PEF for driving persistent carriers separation. This phenomenon is also observed in the oxygen vacancy decorated ferroelectrics.⁷ Notably, the increased maximum polarization in both BTOP and BTOPAu samples compared to BTO and BTOAu₃ is a direct result of the Corona poling pre-switching the domains within the samples, leading to a uniform direction of domains. Additionally, the P_r of BTOPAu is slightly higher than that of BTOAu₃, which is attributed to that different Au structures exhibit varying degrees of this effect.”

We sincerely apologize for the inaccuracies in the data presented in the original manuscript and highly appreciate your valuable comment!

Figure R21 (Fig. 4d). Electric hysteresis loop of BTO, BTOP, BTOAu₃ and BTOPAu (under 10 kV·cm⁻¹).

Figure R22. Electric hysteresis loop of BTO, BTOP, BTOAu₃ and BTOPAu (under 10 kV·cm⁻¹).

5. In Figure 4d, why does not the polarization maximum of BTOP and BTOPAu occur under the largest electric fields of 20 kV cm⁻¹? Even the remnant polarization (zero electric field) is larger than those under some electric fields.

Response: Thank you for your thoughtful comment. We deeply regret for the oversight that there were problems with the sample density and equipment issues during the initial testing, leading to leakage problems during the testing process. Thus, we repeated the experiment, and the new electric hysteresis loops showed that the polarization maximum of both BTOP and BTOPAu samples occur under the largest electric fields. Now, the updated electric hysteresis loops data and the corresponding

analysis were provided in the revised manuscript. Thanks for this value comment again!

6. From the theoretical calculation, the authors revealed the stable formation of the Au array when a voltage is applied. But what the minimum magnitude of the voltage should have? If only talking about the voltage, the BTO with ferroelectricity may also meet this condition. But apparently, the experimental results show BTO cannot form the 1D SAs arrays. Thus, only considering the voltage may not be sufficient to explain the stable formation of 1D SAs arrays.

Response: Thanks for this constructive comment. We have updated the DFT calculation, because the errors in DFT associated with electron delocalization, which is especially acute for strongly correlated systems (Himmetoglu, et al, International Journal of Quantum Chemistry, 114.1 (2014): 14-49), such as the BTO Au considered here. This leads to inaccurate results on energies. These calculations have been re-conducted at the DFT+U level of theory, to conform to the bare minimum of standards in the field. From the updated results, it seems that the BTO can form the 1D SAs arrays in theory, but the formation energy is only -0.20 eV, which is much higher than BTOPAu (-18.25 eV) (Fig. 5a). This difference underscores the highly spontaneous and enduring nature of 1D Au SAs array formation when subjected to an applied voltage.

In the synthetic process, Corona poling was employed to destroy the surface screening field of BTO. This strategic approach led to the establishment of well-ordered anchoring sites conducive to the development of the 1D SAs structure. Despite BTO's inherent ferroelectricity, the prevailing domain orientation is chaotic, with charges and charged ions binding to the surface dipole sites, thus disrupting the orderly anchoring sites. However, the Corona poling treatment breaks the surface screening field and aligns the domains, which provides the ordered anchoring sites. The difference between BTO and BTOP is Corona poling, thus we applied the voltage to simulate the Au loading on BTO and BTOP surface. We cannot totally keep the same condition between experiment and theory, but we still try our best to narrow the

gap between them and provide multiple methods to disclose the mechanism.

Figure R23 (Fig. 5a). Theoretically calculated formation energy of isolated Au atom on $\text{Bi}_4\text{Ti}_3\text{O}_{12}$, Au array on $\text{Bi}_4\text{Ti}_3\text{O}_{12}$ without and with applied voltage.

7. In addition, the authors provide XANES and EXAFS data to support the formation of SACs. However, the explanation of X-ray absorption spectra (including XANES and EXAFS parts) lacks clarity and precision. Since the formation of SACs is the most critical issue in this work, several issues that need to be addressed are as follows: (i) How do the authors normalize the XANES of Au L_3 -edge? The normalization appears to be inadequately performed.

Response: Thank you for this constructive suggestion. We have updated XAS data in revised manuscript for better support the formation of Au SAs, including XANES data normalization and EXAFS data fitting (Fig. 2j, Supplementary Fig. 6 and Supplementary Table 1). Data reduction, data analysis, and EXAFS fitting were performed and analyzed with the Athena and Artemis programs of the Demeter data analysis packages that utilizes the FEFF6 program to fit the EXAFS data.^{8,9} A linear function was subtracted from the pre-edge region, then the edge jump was normalized using Athena software. The $\chi(k)$ data were isolated by subtracting a smooth, third-order polynomial approximating the absorption background of an isolated atom. The k^3 -weighted $\chi(k)$ data were Fourier transformed after applying

a HanFeng window function ($\Delta k = 1.0$). For EXAFS modeling, The global amplitude EXAFS (CN , R , σ^2 and ΔE_0) were obtained by nonlinear fitting, with least-squares refinement, of the EXAFS equation to the Fourier-transformed data in R -space, using Artemis software, EXAFS of the Au foil are fitted and the obtained amplitude reduction factor S_0^2 value (0.856) was set in the EXAFS analysis to determine the coordination numbers (CNs) in the Au-O and Au-Au scattering path in sample. The normalization process is based on the part of 4.1. normalization guide from Athena Users' Guide “<http://bruceravel.github.io/demeter/#about>” and the parameters has been showed in the following Figure R25:

Figure R24 (Fig. 2j). Au L_3 -edge normalized XANES spectra of BTOAu₃, BTOPAu and references.

Figure R25 (Supplementary Fig. 6). Fitting results of **a-d** the k^3 -weighted FT spectrum in R

space and $\mathbf{e}\text{-h } k^3\chi$ data in K space of BTOAu, BTOAu₃ and references at Au L_3 -edge.

Figure R26. The normalized parameters of **a** BTOAu₃, **b** BTOAu, **c** Au foil and **d** Au₂O₃.

Table R4 (Supplementary Table 1). EXAFS fitting parameters at the Au L_3 -edge for samples.

Sample	Shell	CN^a	$R(\text{\AA})^b$	$\sigma^2(\text{\AA}^2)^c$	$\Delta E_0(\text{eV})^d$	R factor
Au-foil	Au-Au	8*	2.845±0.019	0.0151	6.7	0.0032
	Au-Au	6*	2.855±0.009	0.0050	3.1	
Au ₂ O ₃	Au-O	4.1±0.5	1.986±0.001	0.0054	10.6	0.0124
BTOAu ₃	Au-O	3.8±0.2	1.964±0.007	0.0074	7.4	0.0098
BTOAu	Au-O	1.9±0.1	2.035±0.001	0.0016	13.1	0.0180
	Au-Au	4.4±0.9	3.524±0.001	0.0010	3.2	

^a CN , coordination number; ^b R , the distance to the neighboring atom; ^c σ^2 , the Mean Square Relative Displacement (MSRD); ^d ΔE_0 , inner potential correction; R factor indicates the goodness of the fit. S_0^2 was fixed to 0.856, according to the experimental EXAFS fit of Au foil by fixing CN as the known crystallographic value. * This value was fixed during EXAFS fitting, based on the known structure of Au. Fitting range: $3.0 \leq k (\text{\AA}^{-1}) \leq 12.0$ and $1.0 \leq R (\text{\AA}) \leq 3.3$ (Au foil); $3.0 \leq k (\text{\AA}^{-1}) \leq 12.0$ and $1.0 \leq R (\text{\AA}) \leq 2.0$ (Au₂O₃); $1.5 \leq k (\text{\AA}^{-1}) \leq 8.0$ and $1.0 \leq R (\text{\AA}) \leq 2.2$ (BTOAu₃); $2.5 \leq k (\text{\AA}^{-1}) \leq 7.8$ and $1.1 \leq R (\text{\AA}) \leq 3.2$ (BTOAu). A reasonable range

of EXAFS fitting parameters: $0.700 < S_0^2 < 1.000$; $CN > 0$; $\sigma^2 > 0 \text{ \AA}^2$; $|\Delta E_0| < 15 \text{ eV}$; $R \text{ factor} < 0.02$.

(ii) The authors may check carefully if there is a small feature presented at about 11900-11910 eV for BTOAu₃ and BTOPAu. Any interpretation for this small feature?

Response: Thank you for this suggestion. The XANES data around 11900-11910 eV has been zoomed in, and there is no obvious peak in this region (Figure R27). The observed fluctuations in the curve can be attributed to the data with relatively low quality, likely due to the insufficient Au content present, which consequently prevents the emergence of characteristic peaks in the sample. This phenomenon bears similarity to prior research work (Figure R28).⁵

Figure R27. a Au L₃-edge normalized XANES spectra of BTOAu₃, BTOPAu and references. b Enlarged view of red dashed box in a.

Figure R28. Au L_3 -edge XANES spectra of 6 mM TBA[AuCl₂] in DCB (sample #1 in Table1) and 1 mM TBA[AuCl₂] in toluene (sample#2). Permission from ref.^[5].

(iii) How can the authors suggest Au SAs on BTOAu₃ and BTOPAu have valence state between Au⁰ and Au³⁺? The author should explain how they obtain the valence state. It seems the authors did not use the common method (energy of absorption edge or peak maximum) to determine the valence state.

Response: Thank you for this kind suggestion. We have considered about the white line intensity (peak maximum) of Au L_3 -edge to obtain the valence state of Au SAs on BTOAu₃ and BTOPAu. For the L -edge, the white line intensity can reflect the valence state of element, as shown in the above Figure R23. The white line intensity of BTOAu₃ and BTOPAu are between Au foil and Au₂O₃ reference, and thus it can be concluded that the valence state of Au SAs BTOAu₃ and BTOPAu are between Au⁰ and Au³⁺.

(iv) The authors' claim that Au SAs are coordinated with O atoms requires further support. It is recommended that the authors provide the EXAFS fitting results to back up their claim. Without presenting the EXAFS fitting results, it is difficult to justify the coordination environment of Au SAs with O atoms.

Response: Thank you for this important comment. We apologize that the EXAFS fitting data has not provided completely before, and thus we have updated and supplemented the fitting parameters to support the coordination environment of Au SAs (Fig. 2j, Supplementary Fig. 6 and Supplementary Table 1).

(v) Again, why can the authors claim that the second shell observed in BTOPAu is owing to Au-Au coordination without presenting EXAFS fitting results?

Response: Thanks for your value comment. We have made significant improvements to the fitting curves and parameters in the revised manuscript. The updated data, along with the fitting results, are now presented in Fig. 2j, Supplementary Fig. 6 and Supplementary Table 1. There is a small peak around 2.90 Å in BTOPAu,

corresponding to the second shell Au-Au coordination. The fitting parameters from Table R1 indicate that the distance of the Au-Au is around $3.524 \pm 0.001 \text{ \AA}$ in R space, supporting our claim that the second shell Au-Au coordination arises from the Au SAs array structure. Additionally, the $\sigma^2(\text{\AA}^2)^c$, $\Delta E_0(\text{eV})$ and R factor reflect the fitting error, both of them showing a reasonable fitting. Thus, we claimed that the second shell of Au-Au coordination is observed in BTOAu.

(vi) The authors mentioned, "Best-fit parameters extracted from the Au L_3 -edge FT-EXAFS...". However, no fitting parameters are provided in the manuscript, making it difficult to justify whether the fitting is reasonable or not. Following up the above point, the statement "Best-fit parameters...matches the density functional theory"... How close they are and how they can be compared are not addressed here.

Response: Thanks for this constructive comment. We acknowledge that the fitting in the initial manuscript was not perfect and somewhat rough. As a result, we have made significant improvements to the fitting curves and parameters in the revised manuscript. The updated data is now presented in Fig. 2j, Supplementary Fig. 6 and Supplementary Table 1.

We have conducted a detailed analysis of the Au SAs coordination and configuration on BTOAu₃ and BTOAu. The fitting parameters show the first-shell of Au is Au-O structure on both BTOAu₃ and BTOAu, and the second-shell of Au is Au-Au on BTOAu, where the fitting parameters show a reasonable fitting (fitting range: $3.0 \leq k (\text{\AA}^{-1}) \leq 12.0$ and $1.0 \leq R (\text{\AA}) \leq 3.3$ (Au foil); $3.0 \leq k (\text{\AA}^{-1}) \leq 12.0$ and $1.0 \leq R (\text{\AA}) \leq 2.0$ (Au₂O₃); $1.5 \leq k (\text{\AA}^{-1}) \leq 8.0$ and $1.0 \leq R (\text{\AA}) \leq 2.2$ (BTOAu₃); $2.5 \leq k (\text{\AA}^{-1}) \leq 7.8$ and $1.1 \leq R (\text{\AA}) \leq 3.2$ (BTOAu). A reasonable range of EXAFS fitting parameters: $0.700 < S_0^2 < 1.000$; $CN > 0$; $\sigma^2 > 0 \text{ \AA}^2$; $|\Delta E_0| < 15 \text{ eV}$; $R \text{ factor} < 0.02$.) Thus, we created a rational model for DFT calculations based on the XAS results. These efforts have enabled us to obtain a more comprehensive understanding of the different Au SAs structure in our work.

We apologize for any confusion caused by the previous depiction in the original

manuscript. In the revised manuscript, we have made the necessary adjustments to provide a clearer and more accurate statement. The revised depiction now reads as follows:

“The Fourier-transformed EXAFS (FT-EXAFS) spectra and fitting parameters indicate that Au SAs is coordinated with O atoms on the surface of BTOAu₃ and BTOPAu (Fig. 2j; Supplementary Fig. 6 and Table 1). The distinct Au-O coordination is in stark contrast with the metallic Au-Au coordination observed in Au foil. A small peak around 2.90 Å, corresponding to the second shell Au-Au coordination, is observed in BTOPAu, which is proved by the fitting analysis, confirming the formation of Au-O-Au structure in Au SAs array structure (Fig. 2k).”

We sincerely appreciate your constructive and important suggestion, which has significantly enhanced the quality of our work.

(vii) The fitting data for BTOPAu does not match the experimental data, casting doubt on the presence of a second shell Au-Au.

Response: Thank you for this suggestion. We acknowledge that the fitting in the initial manuscript was not perfect and somewhat rough. As a result, we have made significant improvements to the fitting curves and parameters in the revised manuscript. Thanks very much again for all your comments and suggestions above.

References

- 1 Yuan, Y. *et al.* Reduction mechanism of Au metal ions into Au nanoparticles on molybdenum disulfide. *Nanoscale* **11**, 9488 (2019).
- 2 Kim, J. *et al.* Enhanced electrocatalytic properties of transition-metal dichalcogenides sheets by spontaneous gold nanoparticle decoration. *J. Phys. Chem. Lett.* **4**, 1227–1232 (2013).
- 3 Xu, R. Y. *et al.* Preparation of single-atom palladium catalysts with high photocatalytic hydrogen production performance by means of photochemical reactions conducted with frozen precursor solutions. *J. Mater. Chem. A* **11**, 11202–11209 (2023).
- 4 Lábár, L. J. *et al.* Consistent indexing of a (set of) single crystal SAED pattern(s) with the ProcessDiffraction program. *Ultramicroscopy* **103**, 237–249 (2005).
- 5 Chang, S. Y. *et al.* Structure and bonding in Au(I) chloride species: a critical examination of X-ray absorption spectroscopy (XAS) data. *RSC Adv.* **5**, 6912 (2015).
- 6 Bollmann, J. *et al.* Local structure of gold impurities in silicon determined by EXAFS. *Physica B* **376–377**, 57–60 (2006).
- 7 Dong, L. *et al.* Selective hydrogenolysis of 5-hydroxymethylfurfural to 5-methylfurfural over Au/TiO₂. *Appl. Catal., B* **335**, 122893 (2023).
- 8 Ravel, B. *et al.* ATHENA, ARTEMIS, HEPHAESTUS: data analysis for X-ray absorption spectroscopy using IFEFFIT, *J. Synchrotron Radiat.* **12**, 537–541 (2005).
- 9 Zabinsky, S. I. *et al.* Multiple-scattering calculations of X-ray-absorption spectra. *Phys. Rev. B* **52**, 2995–3009 (1995).

REVIEWER COMMENTS

Reviewer #1 (Remarks to the Author):

The authors have responded all my concerns. I recommend that it is published in its current revision.

Reviewer #2 (Remarks to the Author):

I am satisfied with the revisions and thus recommend publishing this work.

Reviewer #4 (Remarks to the Author):

Unfortunately, the authors avoided answering the most critical question in the revised manuscript, which is the “coverage of the Au array”. If the significant enhancement of PC performance results from the aligned Au array, the coverage is very important. If the samples with aligned Au array and without Au array did not have a similar /or comparable coverage, the comparison for the enhanced performance is meaningless. In particular, from the large-area TEM image in the R20, you can see the coverage is not large. How come such a small coverage causes such a large enhancement??

However, the authors have provided only a general response, stating, 'As the coverage is difficult to calculate and not entirely accurate, we have chosen not to include this specific coverage data in the revised manuscript. Thanks for your understanding!' Making such a significant claim without scientific evidence leaves room for skepticism. It is necessary that the authors address this issue, as it plays a crucial role in the overall validity of their conclusion.

In conclusion, I remain unconvinced by the authors' findings until they provide the necessary data regarding coverage. Without this data, the conclusion lacks the necessary support, which is the key for this work.

Reviewer #4:

Unfortunately, the authors avoided answering the most critical question in the revised manuscript, which is the “coverage of the Au array”. If the significant enhancement of PC performance results from the aligned Au array, the coverage is very important. If the samples with aligned Au array and without Au array did not have a similar /or comparable coverage, the comparison for the enhanced performance is meaningless. In particular, from the large-area TEM image in the R20, you can see the coverage is not large. How come such a small coverage causes such a large enhancement??

However, the authors have provided only a general response, stating, 'As the coverage is difficult to calculate and not entirely accurate, we have chosen not to include this specific coverage data in the revised manuscript. Thanks for your understanding!' Making such a significant claim without scientific evidence leaves room for skepticism. It is necessary that the authors address this issue, as it plays a crucial role in the overall validity of their conclusion.

In conclusion, I remain unconvinced by the authors' findings until they provide the necessary data regarding coverage. Without this data, the conclusion lacks the necessary support, which is the key for this work.

Response: We very appreciate this significant and constructive comment! We deeply agree that the Au atom coverage is imperative for a thorough analysis and comparison of the enhanced photocatalytic performance. We sincerely regret our previous misinterpretation of Comment 2 during the last round of revision. For the Comment “2. The authors show the TEM image to support the formation of 1D Au array. What is the large area image of the sample before and after poling?? Since the PC performance was measured under a large area condition, the distribution of 1D Au array, or the coverage of 1D Au array should be provided”, we initially misinterpreted that we should provide the large area image to display the distribution of 1D Au array more obviously. We express our apology again.

Now, we have determined the Au atom coverage for BTOAu₃ and BTOPAu in order to better understand the performance difference of photocatalysts through the following formula:

$$C_{Au} = \frac{N_{Au} \times S_{Au}}{S}$$

Where the C_{Au} is the Au atom coverage, the N_{Au} represents the number of Au atom, S_{Au} is the cross-sectional area of Au and the S is the surface area of substrate. The ICP result reveals the Au loading is 0.27 wt% in BTOAu₃ and 0.30 wt% in BTOPAu (**Supplementary Table 1**). And the BET analysis result indicates the specific surface area of BTOAu₃ and BTOPAu is 11.74 m²/g and 12.25 m²/g, respectively. Consequently, the C_{Au} is determined to be around 6.13% and 6.51% for BTOAu₃ and BTOPAu, respectively. It demonstrates that BTOAu₃ and BTOPAu have comparable Au atom coverage.

Furthermore, for disclosing the enhanced performance is mainly from the Au array permutation, we calculated the ratio of Au array in BTOPAu by combining the DFT calculation (**Fig. R1**) and XANES analysis (valence state and coordination number; **Fig. R2**), which indicates that the Au array is at least 80% pointing its predominant permutation in BTOPAu. The quantitative analysis of Au atom distribution through statistical analysis of the HADDF-STEM images also confirm the above results (**Fig. R3, Supplementary Fig. 6**). **Figure R3a** illustrates a total of 25 discernible bright dots in BTOAu₃, each indicative of an Au atom, which implies the C_{Au} is ~ 6.24% with isolated Au ratio of ~ 100%. In **Fig. R3b**, there are 248 bright dots identified as Au atoms in BTOPAu, with 38 of them being isolated Au atoms and the remaining 210 dots belonging to the Au array, which shows that the C_{Au} is ~ 6.65% and the ratio of Au array is 84.68%. Therefore, it can be concluded that the photocatalytic performance difference between BTOAu₃ and BTOPAu primarily arises from the distinct configurations of Au single atoms (SAs).

Fig. R1 Bader charge transfer of Au atom on **a** BTOAu₃ and **b** BTOPAu model. The charge transfer calculation reveals that the loss charge of Au atom are + 0.824 eV and + 0.553 eV in BTOAu₃ and BTOPAu, respectively.

Fig. R2 The valence state analysis of Au atom. The valence state analysis indicates that the valence state of Au is + 2.82 and + 2.17 in BTOAu₃ and BTOPAu, respectively. Combining the charge transfer in DFT calculation and assuming the ratio of isolated Au atom is 100% in BTOAu₃, it discloses that the ratio of Au array is at least 80% in BTOPAu.

Fig. R3 (Supplementary Fig. 6 in revised manuscript) The HADDF-STEM images of **a** BTOAu₃ and **b** BTOPAu.

Besides, the advantage of single atom catalysts (SACs) is their ability to achieve high performance with minimal metal atom consumption. This is exemplified by many high-performance SAs with low metal loading, as illustrated in **Table R1**. It is worth noting that, in substrates characterized by ultra-high specific surface area or natural confinement pores, the metal atom can attain elevated loading levels. For example, the loading of Co SAs on polymer-derived hollow N-doped porous carbon spheres with large surface area ($568 \text{ m}^2 \cdot \text{g}^{-1}$) can reach 3.54 wt%.¹ The Zn SAs loading can reach 2.66 wt% on microporous N-doped carbon with a specific surface area of $568 \text{ m}^2 \cdot \text{g}^{-1}$.² Besides, single atom alloys can also accommodate a significant number of single metal atoms, as evidenced by the Pb loading in the Pb₁Cu single atom alloy, which reaches 4.4 wt%.³ However, for the bulk or non-porous substrates such as TiO₂, it claimed a “large loading amount” for Cu loading exceeding 1 wt%; however, Cu nanoparticles (about 2–5 nm) are observed when the loading is approximately 2.57 wt%.⁴ Qin et al. also stated similar viewpoints in their paper titled “Single atoms in photocatalysis: Low loading is good enough”.⁵ Millet et al. reported that a Ni SA content ranging from 1-10 atom% on the surface of MgO is sufficient for CO₂ activation. Samples containing 10 atom% Ni or more exhibit a saturation phenomenon, indicating the limitations in performance with more SAs.⁶ Several studies also suggested that the catalytic

performance of low-loading SAs surpasses that of higher loading counterparts. For instance, the photocatalytic performance of 1.1 wt% Fe SACs was found to be superior to that of 4.8 wt% Fe SACs and the Pt loading contents with 0.075 to 0.11 to 0.16 wt% show similar H₂ generation rate.^{7,8} Hence, it can be concluded that even with a relatively low SA loading and coverage, a large catalytic enhancement can be achieved.

Now, we have supplemented the above new data and corresponding discussions in the revised manuscript, as follows:

“The inductively coupled plasma (ICP) results reveal the Au loading with a content of 0.11, 0.15, 0.27, 0.44 and 0.30 wt% on BTOAu₁, BTOAu₂, BTOAu₃, BTOAu₄ and BTOPAu, respectively (**Supplementary Table 1**). The BET analysis result indicates the specific surface area of BTOAu₃ and BTOPAu is 11.74 m²/g and 12.25 m²/g, respectively. The Au coverage is determined to better assess the performance difference of photocatalysts through the following formula:

$$C_{Au} = \frac{N_{Au} \times S_{Au}}{S}$$

Where the C_{Au} is the Au atom coverage, the N_{Au} represents the number of Au atom, S_{Au} stands for the cross-sectional area of Au and the S denotes the surface area of substrate. Consequently, the C_{Au} is determined for BTOAu₃ and BTOPAu to be around 6.13% and 6.51%, respectively. It demonstrates that BTOAu₃ and BTOPAu have comparable Au atom coverage.”

“Besides, the quantitative analysis on the distribution of Au atom through statistical analysis of the HADDF-STEM images illustrate that the average of C_{Au} is around 6.24% and 6.65% of BTOAu₃ and BTOPAu, respectively, confirming the above ICP results (**Supplementary Fig. 6**).”

We are very thankful for this significant comment once again, which inspire us to obtain more comprehensive and important data, enhancing the quality of the manuscript!

Table R1. The photocatalytic performance comparison of single atoms catalysts.

Substrate	Single atom	Loading	Multiples of performance	Ref.
-----------	-------------	---------	--------------------------	------

improvement				
Bi ₄ Ti ₃ O ₁₂	Au	0.30 wt%	~ 18	This work
FeO _x	Pd	0.05 wt%	~ 7	9
Cs ₂ SnI ₆	Pt	0.12 wt%	17.2	10
ZSM-5-H (Si/Al=105)	Rh	0.32 wt%	~ 3 times than commercial Rh/C	11
Pyrolyzed N-C	Ru	< 0.42 wt%	~ 5 times than commercial Pt/C	12
Carbon dots	Cu	0.44 wt%	25	13
TiO ₂ -001	Pt	0.60 wt%	1156	14
TiO ₂	Cu	0.75 wt%	34	15
Bi ₃ O ₄ Br	Co	0.80 wt%	32	16
UiO-66-NH ₂	Pt	1.35 wt%	~116	17
TiO ₂	Cu	1.5 wt%	24.2	4
FAPbBr _{3-x} I _x (FA = CH(NH ₂) ₂)	Pt	1.8 wt%	~ 17	18

References

- Pan, Y. *et al.* Design of single-atom Co–N₅ catalytic site: A robust electrocatalyst for CO₂ reduction with nearly 100% CO selectivity and remarkable stability. *J. Am. Chem. Soc.* **140**, 4218-4221 (2018).
- Han, L. *et al.* Stable and efficient single-atom Zn catalyst for CO₂ reduction to CH₄. *J. Am. Chem. Soc.* **142**, 12563-12567 (2020).
- Zheng, T. *et al.* Copper-catalysed exclusive CO₂ to pure formic acid conversion via single-atom alloying. *Nat. Nanotechnol.* **16**, 1386-1393 (2021).
- Zhang, Y. *et al.* Single-atom Cu anchored catalysts for photocatalytic renewable H₂ production with a quantum efficiency of 56%. *Nat. Commun.* **13**, 58 (2022).
- Qin, S. *et al.* Single atoms in photocatalysis: Low loading is good enough! *ACS Energy Lett.* **8**, 1209-1214 (2023).
- Millet, M.-M. *et al.* Ni single atom catalysts for CO₂ activation. *J. Am. Chem. Soc.* **141**, 2451-2461 (2019).
- Ran, L. *et al.* Engineering single-atom active sites on covalent organic frameworks for boosting CO₂ photoreduction. *J. Am. Chem. Soc.* **144**, 17097-17109 (2022).
- Li, X. *et al.* Single-atom Pt as co-catalyst for enhanced photocatalytic H₂ evolution. *Adv. Mater.* **28**, 2427-2431 (2016).
- Du, P. *et al.* Single-atom-driven dynamic carburization over Pd₁-FeO_x catalyst boosting CO₂ conversion. *Chem* **8**, 3252-3262 (2022).
- Zhou, P. *et al.* Single-atom Pt-I₃ sites on all-inorganic Cs₂SnI₆ perovskite for efficient

- photocatalytic hydrogen production. *Nat. Commun.* **12**, 4412 (2021).
- 11 Sun, Q. *et al.* Zeolite-encaged single-atom rhodium catalysts: Highly-efficient hydrogen generation and shape-selective tandem hydrogenation of nitroarenes. *Angew. Chem. Int. Ed.* **58**, 18570-18576 (2019).
- 12 Xiao, M. *et al.* Engineering energy level of metal center: Ru single-atom site for efficient and durable oxygen reduction catalysis. *J. Am. Chem. Soc.* **141**, 19800-19806 (2019).
- 13 Cai, Y. *et al.* Insights on forming N,O-coordinated Cu single-atom catalysts for electrochemical reduction CO₂ to methane. *Nat. Commun.* **12**, 586 (2021).
- 14 Wei, T., Zhu, Y., Wu, Y., An, X. & Liu, L.-M. Effect of single-atom cocatalysts on the activity of faceted TiO₂ photocatalysts. *Langmuir* **35**, 391-397 (2019).
- 15 Lee, B.-H. *et al.* Reversible and cooperative photoactivation of single-atom Cu/TiO₂ photocatalysts. *Nat. Mater.* **18**, 620-626 (2019).
- 16 Di, J. *et al.* Isolated single atom cobalt in Bi₃O₄Br atomic layers to trigger efficient CO₂ photoreduction. *Nat. Commun.* **10**, 2840 (2019).
- 17 Sui, J. *et al.* A general strategy to immobilize single-atom catalysts in metal-organic frameworks for enhanced photocatalysis. *Adv. Mater.* **34**, 2109203 (2022).
- 18 Wu, Y. *et al.* An organometal halide perovskite supported Pt single-atom photocatalyst for H₂ evolution. *Energy Environ. Sci.* **15**, 1271-1281 (2022).

REVIEWERS' COMMENTS

Reviewer #4 (Remarks to the Author):

The authors have provided the coverage of Au atoms and other related information, which is important for them to make the conclusion for enhanced photocatalytic activity. I suggest this image should be included in the main text part to support the conclusion. The authors have clarified my concern in the revised version.

Reviewer #4 (Remarks to the Author):

The authors have provided the coverage of Au atoms and other related information, which is important for them to make the conclusion for enhanced photocatalytic activity. I suggest this image should be included in the main text part to support the conclusion. The authors have clarified my concern in the revised version.

Response: We sincerely thank the Reviewer for the approval of our manuscript and thank for this kind comment. Now, we have included this image and revised the corresponding discussions in the main text part, as follows:

“Besides, the quantitative analysis on the distribution of Au atom through statistical analysis of the HAADF-STEM images illustrate the average of C_{Au} are around 6.24% and 6.65% of $BTOAu_3$ and $BTOPAu$, respectively, confirming the above ICP results (Fig. 2g, h).

Fig. 2 Structural characterizations of Au SAs on $BTOAu_3$ and $BTOPAu$. a, b HAADF-STEM images of $BTOAu_3$. c, d HAADF-STEM images of $BTOPAu$ (enlarged area from a and b). Line scan HAADF intensity profile of e site A (direction-1 and 2) and f array A (direction-1 and 2) versus distance. g, h The multiple HAADF-STEM images of a $BTOAu_3$ and b

BTOPAu. **i** Au L_3 -edge normalized XANES spectra and **j** Au L_3 -edge EXAFS spectra in R space of BTOAu₃, BTOPAu and references. **k** Fitting results of the k^2 -weighted FT spectrum of BTOPAu at Au L_3 -edge.”